# OpenPros: A Large-Scale Dataset for Limited View Prostate Ultrasound Computed Tomography

**Hanchen Wang**[1,*] **Yixuan Wu**[2,*], **Yinan Feng**[1], **Peng Jin**[3], **Luoyuan Zhang**[1], **Shihang Feng**[1], **James Wiskin**[4], **Baris Turkbey**[5], **Peter A. Pinto**[5], **Bradford J. Wood**[5], **Songting Luo**[6], **Yinpeng Chen**[7], **Emad Boctor**[2], and **Youzuo Lin**[1]

[1]The University of North Carolina at Chapel Hill, [2]Johns Hopkins University,
[3]The Pennsylvania State University, [4]QT Imaging, Inc.,
[5]National Institutes of Health, [6]Iowa State University, [7]Google DeepMind

## Abstract

Prostate cancer is one of the most common and lethal cancers among men, making its early detection critically important. Ultrasound computed tomography (USCT) has emerged as an accessible and cost-effective method that reconstructs quantitative tissue parameters, which can serve as potential biomarkers for malignancy. However, current prostate USCT faces considerable barriers: limited-angle acquisitions due to anatomical constraints, tissue heterogeneity, proximity to organs and bony pelvic structures, and lengthy processing times. The lack of large-scale, anatomically precise datasets significantly hampers the development of high-quality, efficient, and generalizable methods. To address this gap, we introduce OpenPros, the first large-scale benchmark dataset for limited-angle prostate USCT, designed to evaluate machine learning algorithms for inverse problems systematically. Our dataset includes over 280,000 paired samples of realistic 2D speed-of-sound (SOS) phantoms and corresponding ultrasound full-waveform data, generated from anatomically accurate 3D digital prostate models derived from 4 real clinical MRI/CT scans and 62 ex vivo prostate specimens with experimental ultrasound measurements, annotated by medical experts. Simulations are conducted under clinically realistic configurations using advanced finite-difference time-domain (FDTD) and Runge-Kutta acoustic wave solvers, both provided as open-source components. Through comprehensive benchmarking, we find that deep learning methods significantly outperform traditional physics-based algorithms in inference efficiency and reconstruction accuracy. However, our results also reveal that current machine learning methods fail to deliver clinically acceptable, high-resolution reconstructions, underscoring critical gaps in generalization, robustness, and uncertainty quantification. By publicly releasing OpenPros, we provide the community with a rigorous benchmark that not only enables fair method comparison but also motivates new advances in physics-informed learning, foundation models for scientific imaging, and uncertainty-aware reconstruction—bridging the gap between academic ML research and real-world clinical deployment. The dataset is publicly accessible at `https://open-pros.github.io/`.

## 1 Introduction

Prostate cancer is the second most common malignancy in men and is one of the leading causes of cancer-related deaths worldwide. One in eight men suffers from it (Radtke & Hadaschik, 2020; Tosoian et al., 2024). Since the 5-year survival rate for prostate cancer patients significantly drops from nearly 100% to approximately 34% once the disease progresses from localized or regional stages to distant metastases (Institute, 2024), early detection of aggressive prostate cancer is of vital

---

*Equal contribution

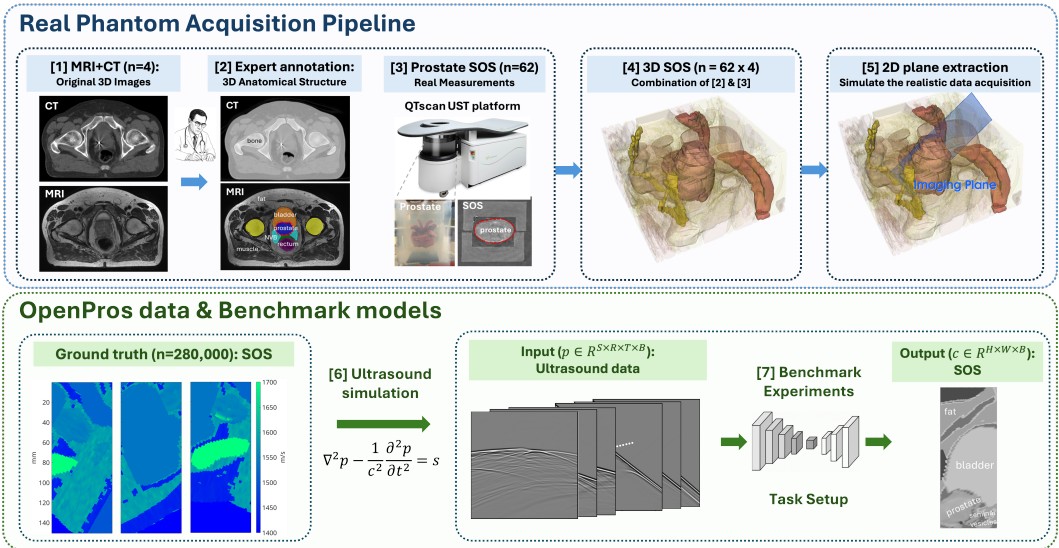

Figure 1: **OPENPROS dataset creation and benchmarking pipeline.** Top panel: Starting from clinical MRI and CT scans, we employ expert annotations to generate detailed 3D anatomical segmentations. We then incorporate real ultrasound speed-of-sound (SOS) measurements from ex vivo prostate samples acquired using the QTscan platform. These are integrated into comprehensive 3D abdominal SOS models. Clinically relevant 2D slices are extracted from these models to simulate limited-angle ultrasound tomography scenarios. Bottom panel: The extracted 2D SOS maps form the ground truth for ultrasound simulations governed by the acoustic wave equation. The resulting simulated ultrasound data are organized into the OPENPROS dataset. We utilize these data to train and benchmark physics-based and deep-learning inversion methods, facilitating the evaluation and development of rapid, clinically relevant SOS reconstruction methods under challenging limited-angle conditions.

importance. Medical imaging plays an essential role in this early detection. Among the available imaging modalities, multiparametric MRI (mpMRI) is currently recognized as the most advanced and accurate imaging tool for detecting and localizing clinically significant prostate cancer. However, the high cost and limited accessibility of mpMRI restrict its widespread adoption, particularly in rural or low-resource settings (De Rooij et al., 2014; Kasivisvanathan et al., 2018).

In contrast, ultrasound imaging is widely accessible, cost-effective, and capable of real-time imaging. Prostate ultrasound is typically performed transrectally, producing B-mode (brightness-mode) images. Although transrectal ultrasound (TRUS) is the clinical standard for routine prostate evaluations and biopsy guidance, it has a sensitivity of only 30%–50% for detecting clinically significant tumors and a specificity of 70%–80% (Beemsterboer et al., 1999; Chen et al., 2016). Studies have further shown that tumors located in the anterior or apical prostate regions are often undetectable with TRUS due to poor soft-tissue contrast and restricted acoustic windows, and TRUS cannot reliably distinguish malignant lesions from benign conditions such as chronic prostatitis (Maričić et al., 2010).

Ultrasound computed tomography (USCT) has emerged as a promising alternative, reconstructing quantitative tissue parameters like speed-of-sound (SOS) and acoustic attenuation that serve as potential biomarkers for malignancy (Wu, 2024; Williams et al., 2021). However, the anatomical constraints of prostate imaging inherently limit the acquisition aperture, creating a challenging *limited-angle condition*. Unlike idealized setups where transducers surround the entire imaging domain, prostate imaging is anatomically restricted to transrectal and transabdominal placements, resulting in sparse and angularly limited data. Traditional physics-based methods typically struggle under these conditions, with slow convergence, severe ill-posedness, and significant reconstruction artifacts (Wang et al., 2025; Gilboy et al., 2020). Developing robust USCT algorithms capable of accurately handling limited-angle data is thus critically needed for clinical prostate imaging.

Furthermore, clinical translation of prostate USCT faces considerable barriers due to the complexity and specialization of current imaging systems. To date, only two USCT systems (SoftVue and QTscan) have received U.S. FDA approval, and both systems focus exclusively on full-angle breast imaging with custom hardware setups unsuited for prostate applications (Sandhu et al., 2015; Malik et al., 2018). These existing systems operate at relatively low frequencies, rely on patient positioning incompatible with prostate imaging, and require hours for reconstruction. Thus, there is an urgent need for efficient, generalizable, and clinically adaptable prostate-specific USCT platforms. Crucially, this advancement depends on the availability of realistic, anatomically precise digital phantoms and datasets, which are currently lacking in the field (Gilboy et al., 2020; Aalamifar et al., 2017).

Additionally, prostate imaging complexity is increased by high tissue heterogeneity and proximity to multiple adjacent organs and bony pelvic structures, invalidating simplified fluid medium assumptions typically used in breast imaging. These factors severely compromise USCT image reconstruction quality, further emphasizing the necessity of specialized prostate-specific datasets.

Recent advances in deep learning, particularly convolutional neural networks (CNNs), have shown potential for overcoming these limitations by learning complex mappings directly from ultrasound data to high-resolution SOS maps (Chugh et al., 2021; Havaei et al., 2017). Data-driven approaches bypass computational bottlenecks encountered by iterative solvers and demonstrate the ability to reconstruct detailed tissue properties even under sparse and noisy acquisition conditions. Instead of hours to days of image reconstruction using physics-based methods and the requirement of expert reading as a follow-up, the relatively short inference time and the automatic analysis enable faster and easier diagnosis for better patient experience. Furthermore, transformer-based architectures have recently demonstrated remarkable performance in medical imaging by effectively modeling long-range spatial dependencies, a feature particularly beneficial for ultrasound tomography due to the extensive spatial interaction of acoustic waves.

Despite these advancements, progress has been significantly hampered by the lack of large-scale, high-fidelity datasets supporting the development, evaluation, and reproducibility of innovative reconstruction algorithms. Existing USCT datasets are typically for breast imaging, which are either synthesized in simulation or derived from real phantoms. Related datasets are listed in Table 1. They exhibit diversity and reflect anatomical realism to a good extent, they are not optimal for developing and benchmarking advancing prostate USCT algorithm. There are also anatomical datasets for the male pelvic region but not for USCT purposes. The commercial anatomy softwares such as Zygote Body and Complete Anatomy provide different pricing options for viewing and downloading, but they generally lack anatomical varieties. Moreover, no publicly available dataset adequately addresses the unique challenges posed by limited-view prostate USCT and the existence of bones in the imaging view while simultaneously providing realistic wave propagation modeling and comprehensive full-waveform data.

Table 1: **Comparison between our OPENPROS and other existing datasets for the male pelvic region or for medical ultrasound computed tomography**. The symbols ✓, ✗, and *NA* indicate that the dataset contains, does not contain, or is not applicable to the corresponding feature, respectively.

| Dataset | Prostate | Acoustic parameters | Actual anatomy | Tissue heterogeneity | Bones | Limited angle | Public | Free access |
|---|---|---|---|---|---|---|---|---|
| **OPENPROS (ours)** | ✓ | ✓ | ✓ | ✓ | ✓ | ✓ | ✓ | ✓ |
| Li *et al.* Li et al. (2021) | ✗ | ✓ | ✗ | ✓ | ✗ | ✗ | ✓ | ✓ |
| Ruiter *et al.* (Ruiter et al., 2018) | ✗ | ✓ | ✗ | ✓ | ✗ | ✗ | ✓ | ✓ |
| OpenWaves (Zeng et al., 2025) | ✗ | ✓ | ✗ | ✓ | ✗ | ✗ | ✓ | ✓ |
| Segars *et al.* (Segars et al., 2010) | ✓ | ✗ | ✓ | *NA* | ✓ | *NA* | ✓ | ✗ |
| The visible human project (Ackerman, 1998) | ✓ | ✗ | ✓ | *NA* | ✓ | *NA* | ✓ | ✓ |
| Zygote Body | ✓ | ✗ | ✓ | *NA* | ✓ | *NA* | ✓ | ✓ |
| Complete Anatomy | ✓ | ✗ | ✓ | *NA* | ✓ | *NA* | ✓ | ✗ |

Motivated by these challenges and the pressing need for prostate USCT, we introduce OPENPROS, the first large-scale dataset specifically designed for limited-angle prostate USCT scenarios. A schematic illustration of the overall pipelines of OPENPROS is shown in Figure 1. Our dataset comprises over 280,000 paired 2D SOS phantoms and ultrasound full-waveform data derived from anatomically realistic 3D prostate models generated from 4 clinical MRI/CT scans and 62 ex vivo prostate specimens with experimentally measured speed-of-sound (SOS), annotated meticulously by clinical experts. OPENPROS serves as a critical benchmark facilitating advances in computational efficiency, limited-angle reconstruction accuracy, rapid clinical adaptability, and comprehensive method comparisons across various imaging conditions, including ray-based, single scattering, high-frequency, and limited-angle scenarios.

In summary, our contributions to the community include:

1. **Large-scale, anatomically realistic benchmark dataset:** The first comprehensive prostate USCT dataset, derived from clinical MRI/CT scans and detailed expert annotations, designed explicitly to address limited-angle imaging conditions.
2. **High-fidelity, publicly available simulation tools:** Advanced finite-difference time-domain (FDTD) and Runge-Kutta implicit iterative acoustic solvers, openly accessible alongside our dataset to facilitate reproducibility and method development.
3. **Comprehensive benchmarking of inversion methods:** Thorough evaluation of physics-based and deep learning methods under realistic limited-angle conditions, including systematic tests of

generalization, robustness, and inference efficiency. These baselines establish clear performance baselines and guide future algorithmic improvements.

The remainder of this paper is structured as follows: In Section 2, we overview the fundamentals of USCT and our task setup. Section 3 details our dataset construction. Section 4 describes benchmarking experiments. In Section 6, we discuss dataset strengths, limitations, and future research directions. Finally, we conclude in Section 7 by summarizing our key contributions and the broader implications of OPENPROS.

## 2 ULTRASOUND COMPUTED TOMOGRAPHY AND FORWARD MODELING

In the context of USCT, the forward problem involves simulating acoustic wave propagation through soft tissues, which is governed by the acoustic wave equation. Assuming an isotropic medium with constant density, the forward modeling equation is given by:

$$\nabla^2 p - \frac{1}{c^2} \frac{\partial^2 p}{\partial t^2} = s, \tag{1}$$

where $\nabla^2 = \frac{\partial^2}{\partial x^2} + \frac{\partial^2}{\partial y^2}$ in 2D, $c(x,y)$ denotes the spatially varying SOS map, $p(x,y,t)$ is the acoustic pressure field, and $s(x,y,t)$ represents the ultrasound source. In our simulations, the source $s$ is prescribed as a controlled ultrasound excitation. Clinically, the primary goal of ultrasound tomography is to reconstruct spatially varying SOS maps from recorded pressure fields, enabling accurate tissue characterization and anomaly detection, such as identifying tumors or lesions.

The forward modeling of ultrasound propagation thus entails computing the pressure field $p$ from a given SOS distribution $c$, represented by the highly nonlinear mapping $p = f(c)$, where $f(\cdot)$ encapsulates the complex wave propagation phenomena defined by Equation (1). In practice, the recorded ultrasound signals form a 4-dimensional tensor $p \in \mathbb{R}^{S \times R \times T \times B}$, where S is the number of sources, $R$ is the number of receivers, T represents the number of time steps, and B denotes the batch dimension. Specifically, in our simulated prostate dataset, we set $S = 20$, $R = 322$, and $T = 1000$. The output SOS maps to be reconstructed are represented as 3-dimensional tensors $c \in \mathbb{R}^{H \times W \times B}$, where $H$ and $W$ represent the spatial height and width dimensions, respectively. In our specific configuration, each SOS map has a spatial resolution of $401 \times 161$ grid points.

Data-driven USCT leverages neural networks to directly approximate the inverse mapping $c = f^{-1}(p)$, as demonstrated in recent studies (Wu & Lin, 2019). Thus, the specific task addressed in this paper is the supervised learning problem, formulated as $\min_\theta \mathbb{E}p, c\,[\mathcal{L}(c, \hat{c})]$, where $\hat{c} = f\theta^{-1}(p)$. Here, $f_\theta^{-1}$ represents a deep neural network parameterized by $\theta$, trained on pairs of simulated ultrasound signals $p$ and corresponding ground-truth SOS maps $c$. The training objective $\mathcal{L}$ typically incorporates quantitative metrics such as Mean Absolute Error (MAE), Root Mean Squared Error (RMSE), Structural Similarity Index Measure (SSIM) and Pearson Correlation Coefficient (PCC).

## 3 OPENPROS DATASET

OPENPROS is the *first* large-scale benchmark dataset explicitly designed to facilitate research in limited-angle prostate ultrasound computed tomography (USCT). It contains anatomically realistic 2D speed-of-sound phantoms, organ segmentation labels, and corresponding simulated ultrasound waveforms derived from detailed 3D digital prostate models. The patient level anatomy ID is named as 3_01, 3_02, 3_03 and 3_04. The prostate level anatomy ID is named as the date of acquisition (in total 62 prostates). Detailed naming strategy can be found in the Appendix A.2. Example data pairs and FDTD simulation code are provided in the supplementary materials

In the following subsections, we first show the basic statistics of our dataset and highlight the related domain interests. We then describe the design strategies of 3D/2D prostate phantoms which maximize the fidelity. At the end, we discuss the ultrasound data simulation setups.

### 3.1 DATASET STATISTICS

OPENPROS consists of 280,000 paired examples of 2D SOS phantoms and ultrasound data, systematically derived from realistic 3D digital models. The essential characteristics and data dimensions are summarized in Table 2.

Table 2: **Dataset summary for the OpenPros USCT dataset**. SOS maps are formatted as (sample × channel (#physical params, SOS here) × depth (vertical) × width (horizontal)); ultrasound data as (sample × channel (#sources) × time × #receivers).

| Dataset | Size | #Train / #Validation / #Test | Ultrasound Data Shape | SOS Map Shape |
|---------|------|------------------------------|----------------------|---------------|
| OpenPros | 6.8 TB | 224K / 28K / 28K | $1140 \times 40 \times 1000 \times 161$ | $1140 \times 1 \times 401 \times 161$ |

OPENPROS supports various critical research topics, including:

**Tissue Interfaces**: Clearly defined interfaces among prostate, bladder, and surrounding tissues, essential for organ boundary delineation and accurate pathology differentiation. Segmentation labels enhance precision in evaluating inversion algorithms.

**Lesion Characterization**: Realistically modeled synthetic lesions (e.g., tumors) introduce SOS discontinuities, challenging algorithms in lesion detection and characterization, critical for diagnostic accuracy.

**Clinical Variability and Realistic Imaging Conditions**: Systematic slicing of high-resolution 3D digital models captures realistic anatomical variability, with advanced finite-difference time-domain (FDTD) simulations reflecting clinical imaging conditions, including limited aperture, acoustic noise, and tissue heterogeneity.

Our sophisticated data generation pipeline, encompassing digital phantom modeling, detailed anatomical labeling, and precise acoustic simulations, significantly enhances the clinical relevance and diversity of the dataset.

### 3.2 3D PROSTATE PHANTOM DESIGN

It is important to note that the 3D digital phantoms were derived from human CT/MRI (Nyholm et al., 2018) and USCT scans of ex vivo prostate specimens (Parikh et al., 2024; Wiskin et al., 2022; Williams et al., 2021). Major organs were annotated by experts using T2-weighted MRI, fat was segmented from T1-weighted MRI, bones were segmented from X-ray CT, and the speed of sound and attenuation of ex vivo prostate samples were measured using a QT scanner (QT Imaging Inc., Novato, California, USA). Speed of sound of other organs

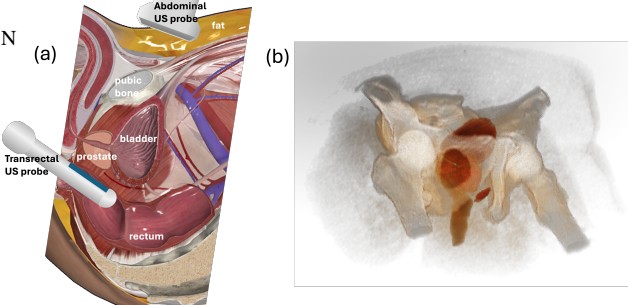

Figure 2: **(a) Anatomical structure and probe placement.** Two probes-abdominal (on the body surface) and transrectal (in the rectum)-are used in our simulation. Image courtesy of Complete Anatomy. **(b) 3D digital SOS phantom.** SOS distribution in the anatomically realistic prostate model.

were acquired from ITIS foundation tissue database (Baumgartner et al., 2024). To best mimic the tissue heterogeneity, we employed Gaussian distributions with given mean values and standard deviations from the tissue database to assign speed of sound in different tissue types. These derivations from human data reduces the reliance on synthetic simulation and maximizes the fidelity of the dataset, especially in the prostate area. Additional details on phantom construction, applications, and open access availability can be found in (Wu et al., 2024). The 3D abdominal anatomical structure and the ultrasound probe placement sketch can be found in Figure 2(a) and the illustration of 3D phantom can be found in Figure 2(b).

### 3.3 2D PROSTATE PHANTOM EXTRACTION

We generate 2D speed-of-sound phantoms by slicing our anatomically realistic 3D prostate volumes under clinically realistic probe configurations. For each phantom, we place one transrectal probe in the rectum and one abdominal probe on the body surface, then sample cross-sections across a range of rotations ($\pm 45°$) and small random perturbations. This process yields 280 K paired 2D SOS maps and corresponding ultrasound waveforms that faithfully capture patient-specific anatomy and limited-angle acquisition variability.

More detailed 2D phantom extraction strategy can be found in Appendix A.1

### 3.4 Ultrasound Data Simulation

We simulate ultrasound wave propagation using a finite-difference time-domain (FDTD) solver based on the 2D acoustic wave equation discussed above. The numerical scheme adopts fourth-order accuracy in space and second-order accuracy in time, offering a reliable trade-off between numerical precision and computational efficiency. This configuration is particularly well-suited for capturing fine-grained wave interactions in heterogeneous prostate tissue environments.

Each simulation is conducted under two acquisition configurations, placing sources and receivers along the top and bottom boundaries of the computational grid. A total of 20 sources (10 at the top and 10 at the bottom, shown as yellow stars in Figure 3) are uniformly distributed along each boundary. For every source, 322 receivers (shown as red dots in Figure 3) are placed across the entire lateral extent of the domain, enabling comprehensive capture of the scattered wavefield. A Ricker wavelet with a 1 MHz peak frequency serves as the excitation pulse, consistent with clinical transducer characteristics. The wavefield is recorded over 1,000 time steps at a sampling interval of $\Delta t = 1 \times 10^{-7}$ seconds, covering a total duration of $100 \ \mu s$. To suppress artificial reflections, 120 grid points of absorbing boundary condition (ABC) are applied to each boundary. Two examples of our simulations and the corresponding SOS maps are shown in Figure 3.

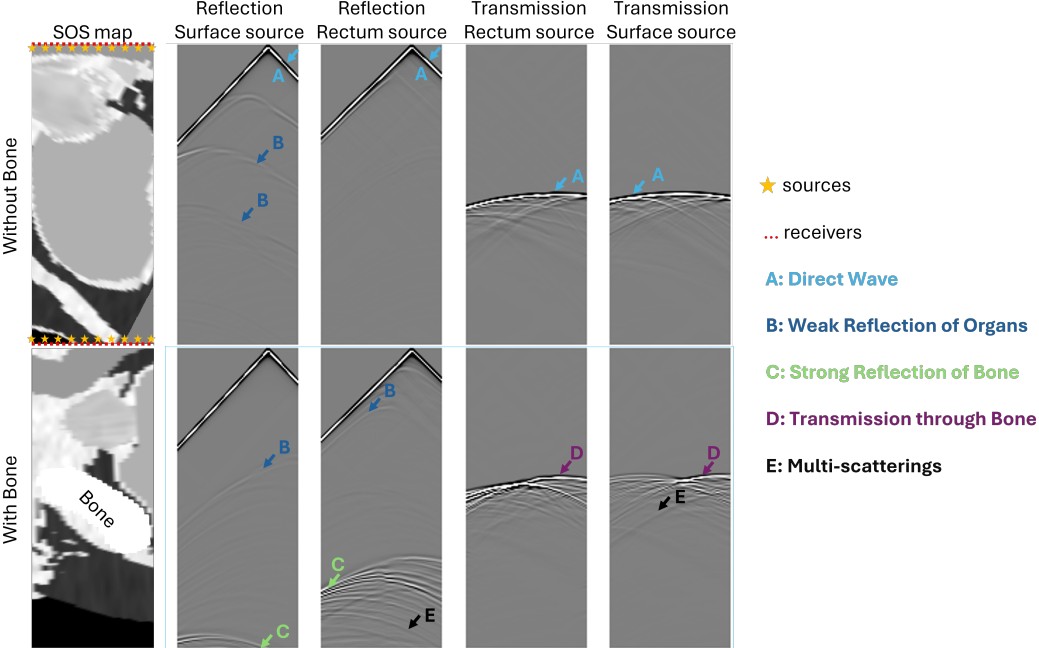

Figure 3: **Examples of simulated ultrasound data and phantoms:** without (top) and with (bottom) bone in the phantoms. We show two example channels of reflections and transmissions with sources (yellow stars) and receivers (red dots) on the two probes. Our PDE solvers can simulate complex and realistic ultrasound wave phenomena, including transmissions, reflections, direct waves, and multi-scatterings.

The spatial discretization of the domain uses a grid spacing of 0.375 mm, resulting in a field of view of approximately 60 mm in width and 150 mm in depth. Each SOS map has a spatial resolution of $401 \times 161$ grid points, corresponding to a physical field of view of 60 mm (lateral) by 150 mm (axial), with uniform grid spacing of 0.375 mm in the lateral and axial directions. These physical dimensions and spacings are kept constant across all examples in the dataset. This configuration mirrors the anatomical scale of the prostate and its surrounding structures. A summary of the physical simulation parameters and the physical meaning of the dimension is shown in Table 3.

Importantly, the simulated ultrasound data for each sample contains 40 distinct channels, organized to reflect practical probe configurations: Channels (0–9) source on body surface, receiver on body surface; (10–19) source on body surface, receiver in rectum; (20–29) source in rectum, receiver in

Table 3: **Physical Meaning of the Prostate USCT Dataset**

| Dataset | Grid Spacing | SOS Map Spatial Size | Source Spacing | Source Line Length | Receiver Spacing | Receiver Line Length | Time Spacing | Recorded Time |
|---|---|---|---|---|---|---|---|---|
| OpenPros | 0.375 mm | 60 mm × 150 mm | 3.75 mm | 60 mm | 0.375 mm | 60 mm | $1 \times 10^{-7}$ s | $100\,\mu s$ |

rectum; and (30–39) source in rectum, receiver on body surface. This setup emulates both conventional transabdominal and transrectal imaging pathways, enabling detailed studies of transmission and reflection across diverse acoustic paths.

## 4 OPENPROS BENCHMARKS

OPENPROS enables systematic investigation of three core questions in limited-angle prostate USCT: (1) **inference efficiency**, (2) **reconstruction accuracy**, and (3) **out-of-distribution (OOD) generalization**. We compare two physics-based baselines, Delay-and-Sum beamforming and multi-stage USCT inversion, against two data-driven models: CNN-based InversionNet (Wu & Lin, 2019) and a Vision Transformer (ViT)-based variant (Dosovitskiy et al., 2020), referred to here as ViT-Inversion. Performance is evaluated using four metrics: mean absolute error (MAE) and root-mean-square error (RMSE) for numerical fidelity, and structural similarity index (SSIM) and Pearson correlation coefficient (PCC) for perceptual and structural alignment. All models are trained and evaluated under identical settings on NVIDIA H100 GPUs.

### 4.1 BENCHMARK METHODS FOR PROSTATE USCT

**Beamforming**, a classical ultrasound imaging method, reconstructs images by aligning and summing received ultrasound echoes according to assumed sound speeds and propagation paths. In this baseline, beamforming serves as a fast, widely-adopted approach for generating initial ultrasound images, highlighting the inherent limitations under restricted-view conditions.

**Physics-based USCT** is performed in a three-stage multi-frequency framework. Starting from a smoothed initial SOS model, we first invert low-frequency data, then mid-frequency data, and finally full-band data. At each stage, synthetic waveforms are generated via our forward operator and compared to observed data; the SOS model is updated by minimizing the waveform misfit.

**InversionNet** (Wu & Lin, 2019) proposed a fully-convolutional network to model the seismic inversion process. With the encoder and the decoder, the network was trained in a supervised scheme by taking 2D (time × # of receivers) seismic data from multiple sources as the input and predicting 2D (depth × length) velocity maps as the output.

**ViT-Inversion**: A Vision Transformer (Dosovitskiy et al., 2020) that

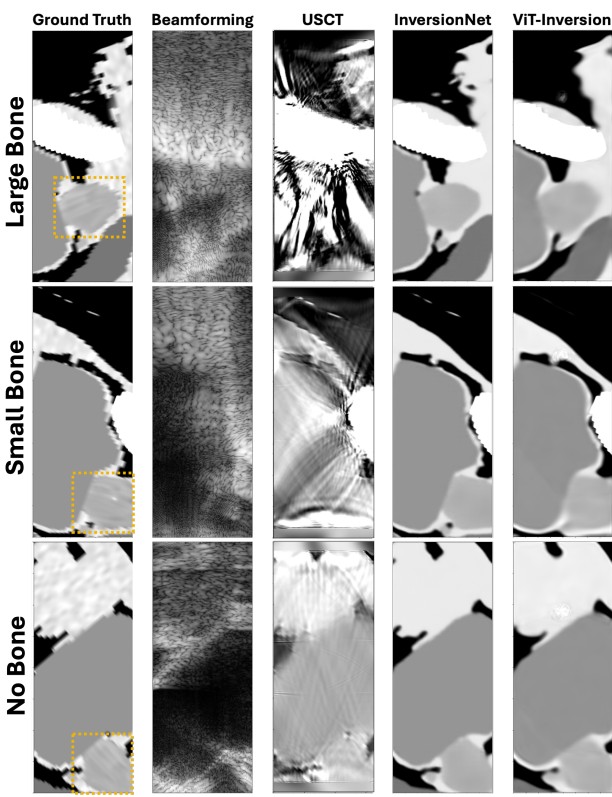

Figure 4: **Benchmark results for limited-angle prostate USCT.** Each column shows a different inversion method on the same phantom: (col 1) ground-truth SOS map; (col 2) Delay-and-Sum beamforming; (col 3) physics-based USCT; (col 4) InversionNet; (col 5) ViT-Inversion. Rows correspond to three representative prostate slices illustrating challenging (top), moderate (middle), and simple (bottom) anatomical scenarios. Zoom-in figures of the prostate region (orange squares) are shown in Figure 5.

partitions the 3D waveform tensor $[S, T, R]$ into spatio-temporal patches, embeds them into tokens, and applies multi-head self-attention to capture long-range wave interactions. A lightweight upsampling CNN refines the patch-wise outputs into full-resolution SOS maps.

**The train/test splits** for the in-distribution (ID) experiments in the section: we randomly partition all available 2D samples at the slice level into training, and test sets (90% / 10%), without enforcing patient exclusivity. As a result, slices from the same 3D anatomy can appear in both train and test sets; this protocol is intentionally designed to measure interpolation performance within the set of anatomies seen during training.

Training configurations and hyperparameters are provided in App. A.4.

## 4.2 RESULT ANALYSIS

Table 4: **Quantitative in-distribution results** (mean $\pm$ std over all test slices).

| Method | MAE↓ | RMSE↓ | SSIM↑ | PCC↑ |
|---|---|---|---|---|
| InversionNet | $0.0074 \pm 0.0037$ | $0.0247 \pm 0.0166$ | $0.9955 \pm 0.0037$ | $0.9851 \pm 0.0262$ |
| ViT-Inversion | $\mathbf{0.0067 \pm 0.0057}$ | $\mathbf{0.0205 \pm 0.0172}$ | $\mathbf{0.9967 \pm 0.0035}$ | $\mathbf{0.9893 \pm 0.0219}$ |

**Quantitative Results**   Table 4 reports MAE, RMSE, SSIM, and PCC for our two learned baselines. Traditional physics-based USCT (not shown) achieves RMSE $\approx 0.16$ and SSIM $\sim 0.90$, leaving substantial room for improvement. In contrast, the learned models reduce RMSE to 0.0297 (InversionNet) and 0.0268 (ViT-Inversion)—about 5–6$\times$ lower than the physics-based baseline—and reach near-perfect structural fidelity (SSIM 0.9877/0.9908). ViT-Inversion is best across all four metrics, followed closely by InversionNet. We also report both the mean and standard deviation of each metric over all test samples. Concretely, for each test slice we compute MAE, MSE, RMSE, PCC, and SSIM between the reconstructed and ground-truth SOS maps, and then aggregate these values across the entire ID test set. Table 4 summarizes these statistics for InversionNet and ViT-Inversion.

**Qualitative Observations** Figure 4 presents three representative prostate slices under challenging (top), moderate (middle), and simple (bottom) anatomical conditions. Delay-and-Sum beamforming yields noisy, low-contrast images incapable of resolving detailed prostate structures. Physics-based USCT significantly reduces these artifacts and better recovers the general gland shape but produces overly blurred images lacking fine anatomical details. Machine learning-based methods, including InversionNet and ViT-Inversion, markedly outperform physics-based

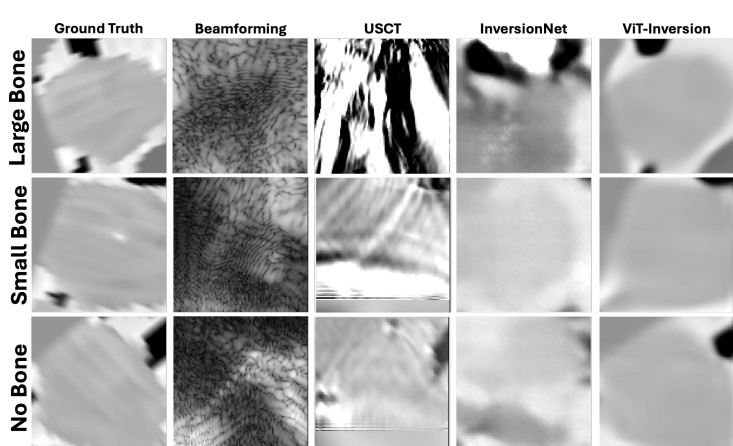

Figure 5: **Zoom-in comparison of prostate regions.** Enlarged views (orange squares in Figure 4) showing detailed reconstruction quality within the prostate region across baseline methods: (col 1) Ground truth; (col 2) Delay-and-Sum beamforming; (col 3) physics-based USCT; (col 4) InversionNet; (col 5) ViT-Inversion. Each row corresponds to the same anatomical scenario as in Figure 4. Note that although the learned methods recover general anatomical shapes more clearly, the fine internal structures and boundaries remain poorly resolved.

USCT in reconstructing the global anatomical structure and boundaries. However, the zoom-in prostate images shown in Figure 5 illustrate that despite better overall shape reconstruction, these learned methods still cannot accurately resolve fine structures within the prostate. The internal prostate structures remain smoothed, and small lesions or detailed boundaries are not distinctly reconstructed, indicating significant room for improvement in imaging resolution and accuracy.

**Inference Efficiency Comparison**   In addition to superior accuracy, data-driven methods offer remarkable computational efficiency suitable for real-time imaging applications, as summarized in

Table 7 (see appendix). Traditional physics-based inversions, such as beamforming and multi-stage USCT, incur significant computational overheads, requiring approximately 4 hours and 24 hours per sample, respectively. In sharp contrast, the data-driven approaches achieve near-instantaneous reconstructions: InversionNet requires only 4.9 milliseconds per sample, while ViT-Inversion completes inference in roughly 8.9 milliseconds due to its transformer architecture. This stark difference highlights the practical feasibility and potential clinical value of learned models in enabling rapid, real-time prostate imaging.

# 5  ABLATION STUDY

**Generalization Tests**   To assess how well our models generalize to truly unseen anatomies, we conducted three out-of-distribution tests using data splits that reflect realistic clinical scenarios: **(1) Patient-Level Generalization:** Train on patients `3_01`, `3_02`, `3_03`, test on an entirely unseen patient `3_04`. **(2) Leave-One-Prostate-Out:** Train on 60 prostates from all four patients except two held-out prostates (i.e., `2022-06-06` & `2022-06-09`); test on those withheld prostates. **(3) Combined Generalization:** Train on 60 prostates drawn only from patients `3_01−3_03`; test on both remaining prostates of patient `3_04`.

Table 5 summarizes MAE, RMSE, SSIM, and PCC for InversionNet and ViT-Inversion under each scenario. In the *patient-level* split, performance drops relative to in-distribution (ID) training, with errors increasing by roughly 3–5×. ViT-Inversion consistently outperforms InversionNet, yielding lower MAE/RMSE and higher SSIM/PCC, which indicates improved generalization ability when encountering anatomies from unseen patients. These results highlight the difficulty of patient-level generalization in limited-angle prostate USCT.

Table 5: **Generalization Test Results.** Evaluation of inversion methods on unseen prostate anatomies. We show (mean ± std) for the InversionNet results and mean for the ViT-Inversion results.

| Scenario | Method | MAE↓ | RMSE↓ | SSIM↑ | PCC↑ |
|---|---|---|---|---|---|
| Patient-level | InversionNet | 0.0322 ± 0.0100 | 0.1010 ± 0.0460 | 0.9399 ± 0.0195 | 0.8271 ± 0.2027 |
| | ViT-Inversion | 0.0276 | 0.0890 | 0.9496 | 0.8689 |
| Prostate-level | InversionNet | 0.0069 ± 0.0064 | 0.0273 ± 0.0157 | 0.9899 ± 0.0067 | 0.9894 ± 0.0174 |
| | ViT-Inversion | 0.0061 | 0.0210 | 0.9934 | 0.9937 |
| Combined | InversionNet | 0.0323 ± 0.0099 | 0.1017 ± 0.0462 | 0.9408 ± 0.0195 | 0.8251 ± 0.2017 |
| | ViT-Inversion | 0.0280 | 0.0916 | 0.9482 | 0.8663 |

By contrast, the *leave-one-prostate-out* split shows minimal and, in some cases, negligible degradation compared to ID. ViT-Inversion achieves slight improvements over InversionNet across all metrics. These results show that intra-patient anatomical variability poses little difficulty for the models, thus generalization across unseen prostates is comparatively straightforward.

The *combined OOD* split mirrors the patient-level challenge: both models exhibit substantial error increases, with ViT-Inversion consistently outperforming InversionNet across metrics. Overall, while intra-patient generalization is well handled, robust patient-agnostic reconstruction under limited-angle conditions remains a significant open challenge.

**Robustness to Input Noise**   We also assess robustness to measurement corruption by adding zero-mean Gaussian noise to the *test* waveforms while keeping all models trained on clean data only; see App. A.6 for full protocol and tables. We sweep $\sigma \in \{0.01, 0.02, 0.05\}$ (roughly $26/23/19\,\mathrm{dB}$ PSNR). Performance decreases monotonically with noise, and ViT-Inversion is consistently more resilient (SSIM $0.987 \to 0.935$) than InversionNet (SSIM $0.944 \to 0.825$). Details and additional metrics (MAE/MSE/RMSE/PCC) are reported in App. A.6.

**Extra Ablation Studies**   We performed extra ablation studies in the appendix, including view-dependent difficulty study A.7, robustness tests of probe placement errors A.8, dataset scaling study A.9, and structure-aware boundary metrics A.10.

## 6    DISCUSSION

Our OPENPROS dataset offers an unprecedented resource for developing limited-angle prostate USCT algorithms. With over 280,000 paired SOS phantoms and ultrasound simulations derived from clinical data, it realistically captures tissue heterogeneity and anatomical constraints of prostate imaging. Our open-source FDTD and Runge-Kutta solvers ensure transparent benchmarking and reproducibility.

However, the dataset currently includes a limited number of patient anatomies, potentially underrepresenting certain anatomical variations. Additionally, we simulate only SOS distributions, omitting other critical acoustic parameters like attenuation and density. Our 2D simulations do not account for three-dimensional propagation and out-of-plane scattering, simplifying some real-world conditions. This 2D, SOS-only setting is a clear simplification compared to a full 3D clinical USCT system, but it preserves the key ill-posedness of limited-angle prostate USCT while remaining computationally tractable. It enables us to generate approximately 280k paired waveform-SOS samples and to systematically compare reconstruction methods at scale. We therefore position OPENPROS as a fast-prototyping benchmark for waveform-to-SOS reconstruction, with the expectation that successful ideas will later be re-evaluated in more complete 3D, multi-parameter models.

Beyond the baseline CNN and ViT models evaluated in this paper, OPENPROS is designed to serve as a testbed for a broader family of reconstruction approaches, including UNet-like encoder–decoder architectures, diffusion and other generative models, and neural operator methods that directly learn PDE solution operators (e.g., (Dai et al., 2023)). The large number of paired waveform–SOS examples and the availability of both ID and OOD evaluation protocols make it possible to systematically study how these different model classes trade off reconstruction fidelity, generalization across anatomies, and computational cost in realistic limited-angle USCT settings.

Future work will expand the patient dataset, introduce multiparametric acoustic maps, and extend simulations to three dimensions. Incorporating clinically relevant pathologies and carefully designed out-of-distribution scenarios will further enhance the robustness and clinical applicability of future USCT solutions.

## 7    CONCLUSION

In this work, we have introduced OPENPROS, the first comprehensive, large-scale benchmark dataset specifically designed for limited-angle prostate ultrasound computed tomography. With over 280,000 expertly annotated 2D speed-of-sound phantoms paired with high-fidelity simulated ultrasound data, OPENPROS facilitates efficient benchmarking of both physics-based and advanced deep learning reconstruction algorithms. Our baseline experiments clearly demonstrate that deep learning approaches significantly outperform conventional physics-based methods in terms of inference speed and image quality. However, critical challenges remain, notably in resolving fine anatomical details within the prostate and achieving robust generalization across unseen anatomies. By making OPENPROS publicly available, we encourage the research community to leverage and expand this foundational resource, ultimately advancing toward clinically viable, high-resolution prostate imaging solutions.

## 8    ACKNOWLEDGEMENT

This work was supported by the University of North Carolina at Chapel Hill School of Data Science and Society through a faculty start-up grant, and by the U.S. National Science Foundation under Award No. 2504439. This work was also supported in part by the NIH Center for Interventional Oncology and the Intramural Research Program of the National Institutes of Health, including the National Cancer Institute and the National Institute of Biomedical Imaging and Bioengineering, through intramural NIH Grants Z1A CL040015 and 1ZIDBC011242, as well as support from the NIBIB Center for Biomedical Engineering Technology Acceleration (BETA Center). Yixuan Wu was supported by the Graduate Partnerships Program, the Fellows Award for Research Excellence, and the Prostate Cancer Research Program Early Investigator Research Award, Congressionally Directed Medical Research Program, U.S. Department of Defense (HT94252410648).

Computational resources were provided by the University of North Carolina at Chapel Hill Information Technology Services Research Computing, the NSF ACCESS program, and the National Center for Supercomputing Applications.

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

## A APPENDIX

Supplementary materials arrangement:

- Section A.1 describes the detailed steps of how 2D phantoms are sliced from the 3D speed-of-sound volumes.
- Section A.2 presents the naming strategy of the OPENPROS dataset.
- Section A.3 details the benchmark evaluation metrics of OPENPROS.
- Section A.4 specifies the baseline models' training configurations and hyper-parameters.
- Section A.5 compares the conventional k-Wave simulation method with the open-sourced OPEN-PROS simulation pipeline.
- Section A.6 analyzes robustness under additive measurement noise.
- Section A.7 investigates view-dependent reconstruction difficulty (bone vs. no-bone propagation paths).
- Section A.8 evaluates robustness to probe placement errors.
- Section A.9 studies dataset scale ablation for learning-based reconstruction.
- Section A.10 introduces structure-aware boundary metrics and gland-level segmentation evaluation.
- Section A.11 describes the usage of large language models in our work.

### A.1 DETAILS OF 2D PHANTOM EXTRACTION

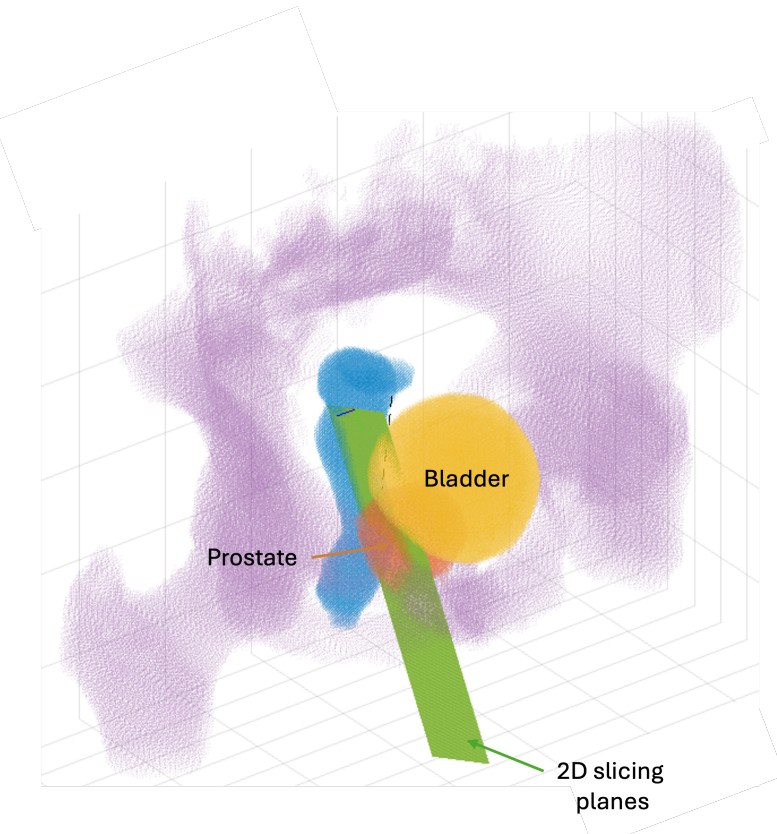

Figure 6: Schematic of our 2D phantom extraction. The green plane indicates the area between transrectal and abdominal probes.

Starting from our 3D prostate volumes (SOS maps and organ masks), we apply the following pipeline to produce each 2D phantom:

1. **Volume loading and isotropic resampling.** We load the segmentation masks and SOS volumes from each patient scan, resample anisotropic voxels onto a uniform 0.375 mm grid, and pad exterior regions with a baseline SOS of 1,500 m/s to emulate coupling gel.

2. **Initial probe placement.** A pair of valid points separated by 6 cm within the segmented rectum defines the transrectal transducer line. The abdominal transducer is then positioned 15 cm anterior along the same axis.

3. **Systematic rotation and translation.** To emulate clinical acquisition angles, we rotate both probe lines jointly from –45° to +45° in 5° increments and translate them within ±1 mm in each Cartesian direction.

4. **Slice extraction.** At each probe configuration, we extract the 2D plane containing both transducer lines. The resulting slice preserves the true anatomical interfaces, tissue heterogeneity, and relative probe geometry.

5. **Random perturbations.** We add a small jitter of ±1° to each rotation angle and ±1 mm to each transducer coordinate to enrich variability.

Altogether, this procedure produces 280,000 unique 2D SOS phantoms with matching ultrasound data under limited-angle conditions. An illustration of the slicing geometry is shown in Figure 6.

## A.2 NAMING OF OPENPROS

The data files follow a structured naming convention: `3_0{i}_P_{Date}_{Category}.npy`, where `{i}` represents patient IDs (1–4), `{Date}` denotes unique prostate sample identifiers, and `{Category}` specifies data types, including ultrasound data (`data`) and SOS maps (`sos`). For example, `3_02_P_2022-06-06_sos.npy` refers to SOS data for patient `3_02` and prostate sample `2022-06-06`.

## A.3 EVALUATION METRICS

To evaluate the performance of the proposed CNN-based reconstruction methods, we employ four quantitative metrics that comprehensively assess numerical accuracy and perceptual similarity between the reconstructed and true SOS images. Here, we denote the true SOS image as $s$, the predicted image as $\hat{s}$, and $N$ as the total number of pixels in each image.

- **Mean Squared Error (MSE)** measures the pixel-wise squared differences:

$$\text{MSE} = \frac{1}{N} \sum_{i=1}^{N} (s_i - \hat{s}_i)^2, \tag{2}$$

where $s_i$ and $\hat{s}_i$ denote the true and predicted SOS values at pixel $i$.

- **Mean Absolute Error (MAE)** calculates the average absolute difference, providing robustness against outliers:

$$\text{MAE} = \frac{1}{N} \sum_{i=1}^{N} |s_i - \hat{s}_i|. \tag{3}$$

- **Structural Similarity Index Measure (SSIM)** assesses the perceptual quality, accounting for luminance, contrast, and structural similarities:

$$\text{SSIM}(s, \hat{s}) = \frac{(2\mu_s \mu_{\hat{s}} + C_1)(2\sigma_{s\hat{s}} + C_2)}{(\mu_s^2 + \mu_{\hat{s}}^2 + C_1)(\sigma_s^2 + \sigma_{\hat{s}}^2 + C_2)}, \tag{4}$$

where $\mu_s, \mu_{\hat{s}}$ denote the means, $\sigma_s, \sigma_{\hat{s}}$ the variances, and $\sigma_{s\hat{s}}$ the covariance between true and predicted images. Constants $C_1$ and $C_2$ stabilize division by weak denominators.

- **Pearson Correlation Coefficient (PCC)** quantifies the linear correlation between true and predicted images, measuring structural consistency:

$$\text{PCC} = \frac{\sum_{i=1}^{N}(s_i - \bar{s})(\hat{s}_i - \bar{\hat{s}})}{\sqrt{\sum_{i=1}^{N}(s_i - \bar{s})^2}\sqrt{\sum_{i=1}^{N}(\hat{s}_i - \bar{\hat{s}})^2}}, \tag{5}$$

where $\bar{s}$ and $\bar{\hat{s}}$ denote the mean pixel values of the true and predicted SOS images, respectively.

## A.4 TRAINING PROCEDURE

All baseline models (InversionNet and ViT-Inversion) were trained using a supervised learning approach on the proposed large-scale OPENPROS dataset. The dataset comprises a total of 280,080 samples, with 255,360 used for training and 27,360 for validation/testing.

For fair comparison, we kept the training settings identical across all methods. We employed the Adam optimizer with an initial learning rate of $10^{-4}$, a batch size of 512, and trained each model for up to 120 epochs. Early stopping was implemented based on validation set performance to prevent overfitting. Table 6 summarizes these training parameters.

Table 6: **Training parameters** consistently used across all baseline methods.

| Total Epochs | Training Samples | Test Samples | Batch Size | Optimizer | Learning Rate |
|---|---|---|---|---|---|
| 240 | 255,360 | 27,360 | 512 | Adam | $10^{-4}$ |

The model sizes and training times for each baseline method are listed in Table 7. All experiments were conducted on NVIDIA H100 GPUs.

Table 7: **Computational cost and model size comparison** for baseline methods on the OPENPROS prostate USCT dataset.

| Method | Training Cost (GPU hour) | Inference Cost (GPU second/sample) | Model Size |
|---|---|---|---|
| Beamforming | N/A | 14400 | N/A |
| USCT | N/A | 86400 | N/A |
| InversionNet | 240 | 0.0049 | 20.45M |
| ViT-Inversion | 128 | 0.0089 | 28.33M |

## A.5 CONVENTIONAL K-WAVE VS. OUR ULTRASOUND SIMULATION ALGORITHMS

Ultrasound imaging methods can be broadly categorized by their simulation paradigms. Conventional TRUS simulations, commonly used for B-mode imaging, rely on signal processing pipelines such as delay-and-sum (DAS) beamforming applied to simulated echoes. These simulations are often implemented using toolboxes like MATLAB k-Wave (Treeby & Cox, 2010), which model acoustic wave propagation using pseudospectral methods and reconstruct images from envelope-detected signals. While fast and widely adopted in the clinical ultrasound community, this approach simplifies underlying physics and often introduces artifacts due to assumptions like homogeneous backgrounds, limited diffraction modeling, or approximate transducer responses.

In contrast, our dataset adopts a physically grounded modeling framework based on the 2D acoustic wave equation. We employ a finite-difference solver with fourth-order spatial and second-order temporal accuracy to simulate full-wave propagation through heterogeneous tissue. This method captures wavefront curvature, multi-path scattering, and fine-grained variations in the SOS map, offering a far more realistic approximation of ultrasound interactions in complex anatomical regions such as the prostate. This fidelity is especially critical under limited-angle acquisition constraints common in prostate imaging, where traditional methods often fail to reconstruct accurate quantitative maps.

Unlike DAS-based methods, our simulation does not rely on beamforming post-processing, allowing it to serve as a foundation for quantitative imaging tasks like USCT. While k-Wave simulations are

computationally efficient for B-mode image formation, our FDTD method remains tractable at scale and better suited for generating ground truth waveforms for learning-based inverse solvers.

Moreover, we provide two forward modeling solvers as part of our open-source release: a finite-difference solver with fourth-order spatial and second-order temporal accuracy, and an alternative Runge-Kutta implicit iterative solver. The former prioritizes accuracy and efficiency in time-domain modeling, while the latter offers enhanced numerical stability and can serve as a basis for future 3D extensions. Both solvers are publicly available with our dataset, supporting reproducibility and extensibility for ultrasound tomography and machine learning research. Additionally, the solvers support execution on both GPU and CPU platforms.

## A.6 NOISE ROBUSTNESS UNDER ADDITIVE MEASUREMENT PERTURBATIONS

**Setup** To assess robustness to measurement corruption, we perturb the test waveforms with zero-mean i.i.d. Gaussian noise and evaluate the pretrained models *without* any fine-tuning (all models were trained only on clean data):

$$\tilde{\mathbf{p}} = \mathbf{p} + \boldsymbol{\epsilon}, \qquad \boldsymbol{\epsilon} \sim \mathcal{N}\left(\mathbf{0}, \sigma^2 \mathbf{I}\right).$$

Noise is applied to the input tensor after the same normalization used during training. We sweep $\sigma \in \{0, 0.01, 0.02, 0.05\}$; for reference, these levels correspond to input PSNRs of approximately 26 dB, 23 dB, and 19 dB, respectively. Reconstructions are compared to the clean ground-truth SOS using MAE, MSE, RMSE, SSIM, and PCC.

**Results** Tables 8 and 9 summarize performance for ViT-Inversion and InversionNet. Both models degrade monotonically as noise increases, but ViT-Inversion is substantially more resilient across all metrics. At $\sigma = 0.01$ ($\approx 26$ dB), ViT-Inversion maintains SSIM $\approx 0.987$ and PCC $\approx 0.982$, while InversionNet drops to SSIM $0.944$ and PCC $0.837$. At the highest noise level ($\sigma = 0.05$, $\approx 19$ dB), ViT-Inversion still preserves moderate structural fidelity (SSIM $0.935$, PCC $0.796$), whereas InversionNet falls to SSIM $0.825$ and PCC $0.388$. These trends suggest that attention-based models better suppress high-frequency perturbations by leveraging longer-range context in the wavefield.

Table 8: **Noise robustness of ViT-Inversion.** Gaussian noise $\mathcal{N}(0, \sigma^2)$ added to test inputs; models trained on clean data only.

| ViT-Inversion | $\sigma = 0$ | $\sigma = 0.01$ | $\sigma = 0.02$ | $\sigma = 0.05$ |
|---|---|---|---|---|
| PSNR (dB) | – | 26 | 23 | 19 |
| MAE | 0.0067 | 0.0084 | 0.0137 | 0.0366 |
| RMSE | 0.0268 | 0.0330 | 0.0521 | 0.1149 |
| PCC | 0.9893 | 0.9824 | 0.9507 | 0.7965 |
| SSIM | 0.9909 | 0.9872 | 0.9759 | 0.9347 |

Table 9: **Noise robustness of InversionNet.** Same protocol as Table 8.

| InversionNet | $\sigma = 0$ | $\sigma = 0.01$ | $\sigma = 0.02$ | $\sigma = 0.05$ |
|---|---|---|---|---|
| PSNR (dB) | – | 26 | 23 | 19 |
| MAE | 0.0074 | 0.0287 | 0.0573 | 0.1228 |
| RMSE | 0.0297 | 0.0964 | 0.1700 | 0.3155 |
| PCC | 0.9851 | 0.8372 | 0.6604 | 0.3881 |
| SSIM | 0.9877 | 0.9437 | 0.8998 | 0.8252 |

**Takeaways and implications** (1) Robustness *without* noise exposure during training is limited—especially for convolution-only models—highlighting the importance of noise-aware data augmentation and/or denoising front-ends. (2) Attention mechanisms appear to confer greater stability to measurement noise in this task. (3) Practical deployment will likely benefit from simple

Table 10: **View-dependent performance of ViT-Inversion** on in-distribution test slices, stratified by bone involvement along the propagation paths. Metrics are reported as means over all slices in each group. Lower is better for MAE/MSE/RMSE; higher is better for PCC/SSIM.

| Metric | No bones (7063 samples) | Small bones (6694 samples) | Big bones (13603 samples) |
|---|---|---|---|
| MAE | 0.0046 | 0.0059 | 0.0082 |
| MSE | $9.63 \times 10^{-5}$ | $4.52 \times 10^{-4}$ | $1.17 \times 10^{-3}$ |
| RMSE | 0.0098 | 0.0213 | 0.0342 |
| PCC | 0.9740 | 0.9884 | 0.9976 |
| SSIM | 0.9921 | 0.9908 | 0.9902 |

defenses such as SNR-matched augmentation, temporal filtering of waveforms, and uncertainty-aware inference; we include these as future baselines in subsequent releases.

### A.7 VIEW-DEPENDENT DIFFICULTY: BONE VS. NO-BONE PATHS

To better understand which acquisition configurations are easier or harder to reconstruct, we stratify the test slices according to whether the dominant ultrasound propagation paths intersect the pelvic bone. Specifically, we group slices into three categories: (i) *no bones*, where rays from the probe to the prostate region do not intersect bone; (ii) *small bones*, where bone intersections occur but occupy a relatively small angular extent; and (iii) *big bones*, where the prostate region is largely shadowed by bone, corresponding to more severe limited-angle geometries.

Table 10 reports the performance of the ViT-Inversion baseline on these three categories in the in-distribution setting. We observe that MAE and RMSE increase as the amount of bone along the propagation paths increases: the *no bones* group yields the lowest errors, the *small bones* group shows intermediate errors, and the *big bones* group exhibits the highest errors. This trend confirms that views with strong bone interference are substantially harder to reconstruct, while bone-free views are comparatively easier. SSIM remains high in all three categories, indicating good overall structural similarity, but amplitude-wise errors are clearly more pronounced in the presence of bone.

### A.8 ROBUSTNESS TO PROBE PLACEMENT ERRORS

In this section, we use experiments to analyze the baseline model's robustness to probe placement errors. To directly address this concern, we regenerated waveform data by perturbing only the abdominal probe position in the forward simulator while keeping the rectal probe fixed. All perturbations use a 2-pixel mismatch on the SOS grid (approximately $0.75\,\mathrm{mm}$ on the FDTD grid). We study three cases:

- **Depth error:** The abdominal sources and receivers are shifted deeper by 2 pixels, with no change applied to the rectal probe.

- **Horizontal mismatch:** The SOS map is padded by 2 pixels on the right for the simulation, and the abdominal probe is shifted laterally by 2 pixels; the rectal probe remains unchanged.

- **Angular mismatch:** The abdominal probe is slightly tilted by lowering its right endpoint by 2 pixels and re-interpolating all element positions along the tilted line; again, the rectal probe is not modified.

We then apply the trained ViT model without any retraining or adaptation, thereby isolating its robustness to geometry mismatch. The quantitative results are shown in Table 11.

Table 11: **Robustness to Probe Placement Perturbations.**

| Mismatch Type | MAE↓ | MSE↓ | RMSE↓ | PCC↑ | SSIM↑ |
|---|---|---|---|---|---|
| Baseline | 0.007374 | 0.000885 | 0.029743 | 0.98509 | 0.98767 |
| Angle | 0.009829 | 0.001807 | 0.042510 | 0.976239 | 0.981784 |
| Depth | 0.013245 | 0.003513 | 0.059269 | 0.959708 | 0.972126 |
| Horizontal | 0.010061 | 0.001608 | 0.040098 | 0.972594 | 0.981417 |

Across all perturbation types, the model maintains high structural similarity and correlation (SSIM $> 0.97$ and PCC $> 0.96$), with depth mismatch producing the largest degradation and horizontal/angle mismatches leading to more moderate changes relative to the baseline. These results indicate that the learned model is reasonably robust to small, controlled errors in probe placement.

Finally, we note that "SOS background shifts" in the sense of perturbing a fixed homogeneous background parameter are not directly applicable in our setting: our USCT inversion directly estimates the full SOS field and does not assume or optimize around a separate homogeneous background model.

## A.9 DATASET SCALE ABLATION

We conduct subset training experiments where InversionNet and the ViT-based model are trained using 10%, 25%, and 50% of the original training set. For both architectures, increasing dataset size consistently improves reconstruction performance: MAE and RMSE decrease, while PCC and SSIM increase.

For example, InversionNet achieves an RMSE reduction from approximately 0.0488 (10%) to 0.0360 (50%), while the ViT model reduces RMSE from approximately 0.0456 (10%) to 0.0258 (50%). These results highlight the importance of large-scale data for learning-based USCT reconstruction.

Detailed quantitative results (mean $\pm$ standard deviation) are reported in Tables 12 and 13.

Table 12: **Subset Training Results for InversionNet** (mean $\pm$ std).

| Metric | 10% | 25% | 50% |
|---|---|---|---|
| MAE | $0.014151 \pm 0.006491$ | $0.013643 \pm 0.018162$ | $0.010478 \pm 0.013810$ |
| MSE | $0.003107 \pm 0.003620$ | $0.003867 \pm 0.022751$ | $0.002462 \pm 0.016337$ |
| RMSE | $0.048756 \pm 0.027011$ | $0.045856 \pm 0.041998$ | $0.035999 \pm 0.034152$ |
| PCC | $0.957021 \pm 0.058652$ | $0.959368 \pm 0.066839$ | $0.973800 \pm 0.047682$ |
| SSIM | $0.971254 \pm 0.012155$ | $0.975261 \pm 0.019335$ | $0.981490 \pm 0.015850$ |

Table 13: **Subset Training Results for ViT-Inversion** (mean $\pm$ std).

| Metric | 10% | 25% | 50% |
|---|---|---|---|
| MAE | $0.013624 \pm 0.006603$ | $0.011839 \pm 0.007811$ | $0.008236 \pm 0.005039$ |
| MSE | $0.002955 \pm 0.004330$ | $0.002255 \pm 0.007821$ | $0.000986 \pm 0.004823$ |
| RMSE | $0.045610 \pm 0.029575$ | $0.039792 \pm 0.025909$ | $0.025761 \pm 0.017956$ |
| PCC | $0.960069 \pm 0.060274$ | $0.968068 \pm 0.053930$ | $0.984177 \pm 0.027592$ |
| SSIM | $0.974953 \pm 0.013378$ | $0.978290 \pm 0.011619$ | $0.987186 \pm 0.008936$ |

## A.10 STRUCTURE-AWARE BOUNDARY METRICS

To complement global image-level metrics, we introduce structure-aware evaluations focused on prostate gland boundaries.

**Boundary-Band Error Metrics** For each test slice, we extract the prostate boundary from the ground-truth mask and define a narrow band of width 5 pixels centered around this boundary. Within this boundary band, we compute MAE, MSE, and RMSE to assess reconstruction accuracy in anatomically critical regions.

For InversionNet, the boundary-band RMSE is $0.0131 \pm 0.0137$, indicating accurate recovery of gland contours. This boundary-focused evaluation provides a clinically meaningful complement to global averages, as it directly measures reconstruction fidelity in regions relevant for gland delineation and downstream diagnostic tasks.

**Gland-Level Segmentation Evaluation** In addition to boundary-band errors, we perform a segmentation-based evaluation on reconstructed SOS maps.

Table 14: **Structure-Aware Boundary Metrics (InversionNet)** computed within a 5-pixel prostate boundary band.

| Metric | Mean | Std |
|--------|----------|----------|
| MAE | 0.005534 | 0.004368 |
| MSE | 0.000359 | 0.000986 |
| RMSE | 0.013097 | 0.013678 |

We follow MedSAM (Ma et al., 2024) and use its public implementation (`https://github.com/bowang-lab/MedSAM`) as a proxy for a strong prostate segmenter. For each test slice in a selected subset, we manually draw a bounding box around the prostate and:

1. Apply MedSAM to the ground-truth SOS map and compute the Dice score against the ground-truth prostate mask.

2. Apply MedSAM to the predicted SOS map and compute the Dice score against the same ground-truth mask.

Because MedSAM is applied identically in both cases, this comparison isolates the degree to which gland-level structural information is preserved in the reconstructed SOS maps.

Table 15: **Gland-Level Segmentation Evaluation Using MedSAM.** Dice scores computed against ground-truth prostate masks.

| Input to MedSAM | Reference Mask | Dice Score |
|-----------------|------------------|------------|
| Ground-Truth SOS | GT Prostate Mask | 0.831982 |
| Predicted SOS | GT Prostate Mask | 0.831345 |

The absolute Dice difference between ground-truth and reconstructed SOS inputs is less than $7 \times 10^{-4}$, indicating negligible degradation in gland-level segmentation performance. This result demonstrates that the reconstructed SOS maps preserve prostate boundary information at a level comparable to ground truth for downstream gland-level tasks.

We emphasize that OPENPROS is designed primarily for quantitative reconstruction accuracy assessment. Accordingly, pixel-wise metrics (MAE, RMSE) and structural similarity metrics (SSIM, PCC) remain essential for comprehensive benchmarking, while the structure-aware metrics reported here provide complementary clinical relevance.

## A.11 USAGE OF LARGE LANGUAGE MODELS (LLMS)

During the preparation of this manuscript, we used a large language model (LLM) to assist with language polishing, structural refinement, and presentation clarity. The LLM provided feedback on phrasing, grammar, and flow, suggested alternatives to reduce redundancy, and generated LaTeX formatting for tables and equations. All scientific ideas, experiments, analyses, and conclusions were conceived, designed, and carried out entirely by the authors.

