# OpenReview forum: "OpenPros: A Large-Scale Dataset for Limited View Prostate Ultrasound Computed Tomography"
_ICLR.cc/2026/Conference — ICLR 2026 Poster_

### Official Review · Reviewer_KYSw · 2025-10-26

**Soundness:** 3
**Presentation:** 3
**Contribution:** 2
**Rating:** 4
**Confidence:** 3

**Summary:**

The authors introduced OPENPROS, the first large-scale benchmark dataset for limited-angle prostate Ultrasound computed tomography, designed to evaluate machine learning algorithms for inverse problems. Baseline comparisons proposed based on physics-based approaches and DL-based approaches. Dataset is open source, and seems like quite clinically meaningful in the specific use case. However, several limitations exists. For example but not limited to, it is hard for me to evaluate if it is truly clinically meaningful because I’m lacking the specific domain knowledge, etc.

**Strengths:**

1. The proposed dataset size is large.
2. The authors stated that this dataset is the first one in their specific use case, could be meaningful.
3. The manuscript is well-written.

**Weaknesses:**

1. More detailed explanation of the relationships in between USTC, 2D SOS phantoms and ultrasound is needed. I cannot understand the clinical problem behind. Many reviewers have expertise in medical imaging, many also have expertise in prostate imaging, but I guess very limit readers and reviewers understand the relationship among them. For example, although I’m familiar with abdominal imaging and related applications, when I looked at the statement “Our dataset comprises over 280,000 paired 2D SOS phantoms and ultrasound full-waveform data derived from anatomically realistic 3D prostate models generated from clinical MRI/CT scans and ex vivo ultrasound measurements, annotated meticulously by clinical experts.”, I don’t know what does it really mean. 280,000 is a very large number in general, but what about in the context of USTC, 2D SOS phantoms, etc. ? What should I focus? I think this part needs to be emphasized since you are proposing a dataset, and it should impress the readers about the quantity of the data, quality of the data and the clinical meaningfulness about the proposal. Now I don’t feel them. Or, maybe it is purely my lack of expertise. If other reviewers have the similar problem or questions, I would like to suggest you to seek for conferences like MICCAI, or journals like IEEE TMI, MIA etc. I do believe that you put lots of effort into the dataset, but it’s hard for me to really measure it.
2. “It contains anatomically realistic 2D speed-of-sound phantoms, organ segmentation labels, and corresponding simulated ultrasound waveforms derived from detailed 3D digital prostate models.” It seems like the data are generated from phantoms not patients, and the ultrasound waveforms were simulated. Therefore, how the study really means in clinical applications might be questionable.
3. “two data-driven models: CNN-based InversionNet Wu & Lin (2019) and a Vision Transformer (ViT)-based variant Dosovitskiy et al. (2020), referred to here as ViT-Inversion.” No image reconstruction methods related to inverse problems like UNet-like or Diffusion-based models (could be other generative models) introduced.

**Questions:**

1. “Our dataset comprises over 280,000 paired 2D SOS phantoms and ultrasound full-waveform data derived from anatomically realistic 3D prostate models generated from clinical MRI/CT scans and ex vivo ultrasound measurements, annotated meticulously by clinical experts.” Have all these patients underwent RP to take the whole prostate out so that you can do the ex vivo measurement? Or it is just ex vivo phantoms measurements?
2. “In our specific configuration, each SOS map has a spatial resolution of 401 × 161 grid points.” Is the physical resolution the same across different SOS maps?
3. Is the simulated ultrasound meaningful enough in your use case in comparison to real patients’ ultrasound?
4. “To best mimic the tissue heterogeneity, we employed Gaussian distributions with given mean values and standard deviations from the tissue database to assign speed of sound in different tissue types.” Do you have any references saying this approach could be reasonable?
5. When the authors introduced the samples, rather than only mention 280,000 slides, it would be better to also mention how many patients got involved in the study and the dataset, since generally you would like to split the train/test set by patients, not by slides to guarantee the independence.

**Details Of Ethics Concerns:**

This is the first time I'm reviewing paper related to open source clinical dataset. I'm not sure if Ethics Concerns are corrected selected or not, but just want to make sure.

---

> ### Author Response · Authors · 2025-11-20
>
> **Q1: Relationship between USCT, 2D SOS phantoms, and ultrasound; unclear clinical problem**
>
> Response:
> We thank the reviewer for this valuable feedback. To improve accessibility for non-USCT experts, we have added:
>
> - A conceptual pipeline figure early in the paper, illustrating:
> 3D clinical anatomy → 3D SOS phantom (with tissue-specific SOS) → 2D slice / SOS map → ultrasound full-waveform simulation → reconstruction of the 2D SOS map.
> - A concise, plain-language explanation in the Introduction and Abstract clarifying that: 1. A 2D SOS phantom is a cross-sectional map of the speed of sound in tissue (in m/s), representing how fast ultrasound waves travel through each pixel. 2. Each sample consists of a ground-truth 2D SOS map and the simulated ultrasound waveforms that would be recorded by transrectal or transabdominal probes placed around the pelvic region. 3.	These paired data are used to evaluate algorithms that invert waveforms back into quantitative tissue property maps, i.e., solving an inverse problem.
> - A clearer explanation of how the 280k samples are generated by combining 4 patient anatomies, 62 ex-vivo prostates, and multiple probe configurations and angles, so that non-USCT readers can understand both the anatomical and acquisition dimensions of variability.
>
> **Q2: Physical resolution of 401*161 SOS grid**
>
> Response:
> Thank you for highlighting this concern. We have explained the label and data dimensions in the original submission Table 2. In the revised Methods section, we now also explicitly state that each SOS map has a resolution of 401 * 161 grid points and give the corresponding physical field of view (lateral and axial) and grid spacing. We also clarify that these physical dimensions and spacings are identical for all examples in the dataset. This makes the physical scale of the images explicit and confirms that it is fixed across anatomies and phantoms.
>
> **Q3: Meaningfulness of simulated ultrasound vs real patient data**
>
> Response:
> Thank you for raising the concern, which have also been mentioned by other two reviewers above. The responses to *kNhg Q8-9* and *ow6B Q4* have answered the question.
>
> We directly address this in the revised Limitations section. We acknowledge that the current dataset is based on simulated waveforms derived from anatomically realistic but still synthetic phantoms, and that this is a simplification compared to real clinical acquisitions. We emphasize that:
>
> - The 3D anatomical backgrounds are derived from actual MRI/CT scans,
> - The SOS values are anchored in ex-vivo measurements of fresh prostate tissue at 37 °C, and
> - The wave propagation is modeled using established time-domain solvers (FDTD).
>
> At the same time, we clearly state that the simulations omit some real-world complexities (e.g., true 3D propagation with full attenuation and scattering) and that transferring models trained solely on simulated data to clinical data will require further validation and adaptation. We present OPENPROS as a realistic but controlled simulation benchmark designed to enable algorithmic progress and systematic evaluation, not as a replacement for clinical trials.

---

> ### Author Response · Authors · 2025-11-20
>
> **Q4: Gaussian tissue heterogeneity modeling**
>
> Response:
> In the revised Methods section, we now include:
>
> - A citation to tissue-property databases and prior parametric phantom models that use Gaussian variability around mean SOS values to represent tissue heterogeneity.
> - A short discussion noting that Gaussian perturbations are a simplified statistical model of heterogeneity, capturing first-order variability but not all higher-order spatial correlations or pathology-specific changes.
>
> We explicitly state that this modeling is a pragmatic compromise that allows us to efficiently generate a diverse library of SOS maps, and we identify more sophisticated heterogeneity models (e.g., learned texture models, pathology-conditioned fields) as a promising direction for future dataset extensions.
>
> **Q5: Number of patients vs 280K slices; splits by patient; scale interpretation**
>
> Response:
> We have made several changes to address this concern:
> 	1.	Patients vs slices:
> Early in the Introduction and in the dataset description, we now clearly state that the dataset is built from four MRI/CT-derived pelvic anatomies and 62 ex-vivo prostates, and that the ≈280k 2D slices are generated by combining these anatomies/prostates with multiple probe configurations and angles.
> 	2.	Splitting strategy:
> We explicitly describe how splits are constructed:
>
> - In-distribution splits use random slice-level partitioning (e.g., 90%/10%), allowing slices from the same anatomy to appear in both train and test sets (interpolation setting).
> - OOD splits enforce anatomy-level independence, with patient-level and prostate-level held-out experiments, so that no slices from a held-out anatomy/prostate appear in the training set.
> 	3.	Scale interpretation:
> We add narrative explaining that the 280k examples represent a trade-off: limited patient-level diversity (4 anatomies) but rich prostate-level and geometric diversity (62 distinct prostates, multiple probe placements, and bone-shadowing patterns). We support this by reporting subset training experiments (10%/25%/50% of the training data), which show that performance improves as more examples are used, indicating that the dataset scale is beneficial for reconstruction models. Experiment details can be found above in *PsHV Q3*.

---

> ### Author Response · Authors · 2025-11-23
>
> Thank you for your thoughtful review and for raising important questions about clinical meaning and clarity. In our rebuttal, we added a clearer explanation and schematic of the full pipeline (USCT → 3D phantoms → 2D SOS maps → simulated waveforms → reconstruction), made the dataset composition and physical resolution explicit, discussed the modeling assumptions (Gaussian tissue heterogeneity, simulated ultrasound vs real patient data), and clarified the intended clinical and ML scope of OPENPROS as a reconstruction benchmark with limited anatomical diversity but large geometric/prostate variability. We will upload the remaining planned experiments and updates during the next two days. If you have any further questions or comments on our answers, we would greatly appreciate your feedback.

---

### Official Review · Reviewer_ow6B · 2025-10-28

**Soundness:** 3
**Presentation:** 4
**Contribution:** 3
**Rating:** 6
**Confidence:** 4

**Summary:**

This paper introduces OPENPROS, a new large-scale (280,000 samples) benchmark dataset for limited-angle prostate Ultrasound Computed Tomography (USCT) . The authors identify a critical gap in prostate cancer imaging: while USCT can reconstruct quantitative tissue biomarkers like Speed-of-Sound (SOS), it is hampered by anatomical constraints (e.g., the pelvic bone) that create a severe limited-angle acquisition problem . The lack of realistic data has hindered the development of robust reconstruction algorithms.

The primary contribution is the dataset itself, which features a high-fidelity data generation pipeline: 3D anatomical models are derived from real clinical MRI/CT scans (from 4 patients), and these are combined with real ex vivo ultrasound SOS measurements (from 62 prostate samples) . This anatomical ground truth is then used with advanced, open-sourced FDTD and Runge-Kutta wave solvers to generate paired SOS maps and full-waveform ultrasound data

The paper also provides a comprehensive benchmark, comparing traditional physics-based methods against deep learning models (InversionNet, ViT-Inversion) . The key findings are: 1) DL methods are faster and more accurate at general reconstruction than physics-based methods . 2) Critically, current DL models fail to resolve fine internal prostate structures and exhibit poor generalization to unseen patient anatomies

**Strengths:**

1. This is the first public, large-scale dataset specifically for limited-angle prostate USCT.

2. The method of creating phantoms by combining real MRI/CT anatomical structures with ex vivo SOS measurements is a key strength, ensuring high anatomical realism

3. Providing the FDTD and Runge-Kutta solvers is a valuable contribution that enhances reproducibility and enables future work.

4. The authors are transparent about the failures of current DL methods, particularly in resolving fine details and in OOD generalization , which correctly frames this as a challenging open problem.

**Weaknesses:**

1. The most significant weakness is that the 280,000 samples are derived from the anatomical structures of only 4 patients. This homogeneity means models may simply overfit to these 4 anatomical configurations, and the "patient-level" OOD test (train on 3, test on 1) is statistically insufficient to make strong claims about generalization.

2. The paper is heavily motivated by cancer detection and claims the dataset includes "synthetic lesions". However, the entire benchmark and results section focuses only on reconstructing general anatomy . The clinical-facing motivation is completely disconnected from the actual technical evaluation.

3. The use of 2D simulation is a major simplification. It ignores real-world 3D propagation and out-of-plane scattering effects, meaning models trained on this data may not be at all applicable to real 3D clinical systems.

4. There is a contradiction in the text regarding the simulation aperture. The text implies a wide aperture ("entire lateral extent" of 150mm), while Table 3 suggests a 60mm length. Both 60mm and 150mm seem much larger than the clinical probes shown in Figure 2a. This suggests the simulation may be providing more data than a real probe, making the problem artificially easier.

**Questions:**

1. The motivation is cancer detection and the dataset claims to have lesions. Why is there no benchmark, metric, or qualitative result related to lesion detection or reconstruction in Section 4?

2. Given the N=4 patient limitation, how can the community be confident that models trained on this data are learning generalizable features of prostate USCT, rather than just overfitting to the 4 specific pelvic bone structures in the dataset?

3. Can you please clarify the actual aperture size used for the 322 receivers? Is it 60mm (Table 3) or 150mm (Section 3.4)? How does this compare to the real-world probes in Figure 2a you aim to model?

---

> ### Author Response · Authors · 2025-11-20
>
> **Q1: Cancer motivation vs lack of lesion/clinical benchmarks**
>
> Response:
> We fully agree that there was a mismatch between the cancer-focused motivation and the scope of the original experiments. In the revised manuscript, we clarify that the current release of OPENPROS aims to benchmark quantitative waveform-to-SOS reconstruction under challenging limited-angle conditions, and not lesion detection.
>
> We explicitly state that:
> - The experiments evaluate global and boundary-focused reconstruction quality (including the new prostate boundary-band metrics),
> - We do not claim that the baseline models achieve clinical lesion-detection performance, and
> - Using OPENPROS for lesion- or cancer-focused tasks will require defining additional labels, tasks, and metrics.
>
> We also note that lesions are inherently present in the ex-vivo prostates and their signatures are encoded in the SOS maps, so improved reconstruction of the gland and its internal structures is a prerequisite for any clinically meaningful lesion assessment. We now frame lesion/cancer detection based on OPENPROS as a future research direction, rather than as an outcome of the current benchmark.
>
> **Q2: Limited N=4 anatomies vs generalization claims; OOD analysis**
>
> Response:
> We acknowledge that n = 4 anatomies is a significant limitation for claiming strong patient-level generalization. In the revised manuscript, we:
>
> 	1.	Reframe generalization claims:
> We position OPENPROS as a benchmark / challenge problem rather than a fully generalizable clinical dataset. We explicitly state that the small number of anatomies limits the strength of patient-level generalization claims and that our OOD results should be interpreted as indicative trends.
>
> 	2.	Report variability in OOD metrics:
> _(We have responded in the above answers to reviewer *kNhg Q4*. Please find more details there.)_
>
> We now summarize OOD performance for InversionNet using mean ± std for multiple OOD settings.
>
> These results are presented in a main-paper OOD table and discussed explicitly. We stress in the text that, due to the small number of anatomies, these variability estimates have limited statistical power and should be viewed as descriptive rather than definitive.
>
> 	3.	Discuss trade-off between anatomical and geometric variability:
> We explain that, despite the limited number of patient anatomies, the combination of 62 ex-vivo prostates and multiple acquisition geometries yields substantial variability in prostate morphology and bone shadowing patterns. We now present subset training experiments (10%/25%/50%) showing that performance improves with more training data, supporting the value of the dataset scale while acknowledging the need for more anatomies in future releases. The experiment details can be found in the above responses to *PsHV Q2*.
>
> **Q3: Aperture size and realism (60 mm vs 150 mm)**
>
> Response:
> We thank the reviewer for catching this inconsistency. In the revised manuscript, we carefully reconcile the description of the aperture size:
>
> - We correct any inconsistent numbers so that the stated aperture in the text matches the value used in the simulations and reported in the tables: "Each SOS map has a spatial resolution of 401 * 161 grid points, corresponding to a physical field of view of 60 mm (lateral) by 150 mm (axial), with uniform grid spacing of 0.375 mm in the lateral and axial directions. These physical dimensions and spacings are kept constant across all examples in the dataset."
>
> - We explicitly compare the simulated aperture length to the physical sizes of the transrectal and transabdominal probes shown in the clinical figures, explaining how the simulated aperture is chosen to approximate the aggregate active aperture of realistic systems under idealized conditions.
>
> We also clarify that our goal is to create a challenging but controlled benchmark that reflects the limited-angle nature of the clinical problem (due to bone and geometry), rather than to replicate every hardware detail exactly. This clarification is now included in the Methods section where we describe the acquisition geometry.
>
> **Q4: 2D SOS-only model; missing out-of-plane effects**
>
> Response:
> We agree that our current forward model is a simplification compared to a full 3D, multi-parameter clinical system. Reviewer *kNhg* also has the concern. We have responded to the concern above in *kNhg Q8-9*. Please find more details there.

---

> ### Author Response · Authors · 2025-11-23
>
> Thank you for your careful review and for highlighting both the strengths and limitations of our work. In our rebuttal, we clarified the benchmark nature of OPENPROS and the limited generalization claims given only 4 anatomies, separated the reconstruction-focused scope from cancer/lesion detection (framing the latter as future work), added more detailed OOD variability analysis and view-dependent difficulty (bone vs no-bone) experiments, and resolved inconsistencies in the description of the acquisition geometry and simulation setup. We plan to provide the remaining planned experiments and corresponding updates within the next two days. Please let us know if you have any questions or suggestions regarding our responses. We very much appreciate your feedback.

---

> > ### Comment · Reviewer_ow6B · 2025-11-26
> >
> > Thanks for providing the additional analysis regarding the questions I raised.

---

### Official Review · Reviewer_PsHV · 2025-11-01

**Soundness:** 4
**Presentation:** 3
**Contribution:** 2
**Rating:** 4
**Confidence:** 4

**Summary:**

The authors present a dataset (OpenPros) for ultrasound computed tomography (USCT) of the prostate that is generated from 4 anatomical MRI+CT images, expert annotations of different anatomical areas, speed-of-sound measurment, and ultrasound simulation using a wave equation. The authors compare OpenPros to other available USCT datasets and include benchmarks for physics-based and data-driven methods that measure (1) inference efficiency, (2) reconstruction accuracy, (3) and out-of-distribution generalization. Overall the paper is well written, most of the methods are explained clearly, and the paper is reasonably well-justified as a dataset contribution to field. However, there is not much clarity around whether whether the simulated dataset is sufficient to serve as a comprehensive machine learning benchmark for image reconstruction e.g. are 280K examples are derived from just 4 patients, which limits the effectiveness of the presented benchmarks.

**Strengths:**

- The paper is very well written and justified
- The paper does a good job of designing tasks for the dataset that could potentially determine its efficacy. The areas of (1) inference efficiency, (2) reconstruction accuracy, (3) and out-of-distribution generalization are all interesting and important.
- The paper does a good job of comparing to the USCT literature, including deep learning methods for image reconstruction.

**Weaknesses:**

- The dataset size is a severe limitation in my opinion. It is not at all clear why 280K number of samples are needed for machine learning training. Ablation experiments controlling the number of generated samples could help here in showing the increased benefit from more samples, justifying the approach.
- The metrics in Table 5 are not given enough context. Figure 5 offers some insight into potential benefits of data-driven methods, but this could be better quantified using the annotations.
- Additional analysis of the types of representations learned or exploited by models mapping the prostate tissue seems to be important for the community, but not included in the paper. Even an assessment of which views are easily reconstructed, and which fail, would make a significant difference in the quality of the paper and contribution for the ICLR community.

**Questions:**

- I am actually unsure about the suitability of this paper for ICLR. Why do the authors think this is important for the ICLR community (typically focused on neural network representations or similar)? To be clear, I think this paper is valuable for the broader machine learning community, but seems better suited for IEEE TCI, APL Bioengineering, or even CVPR or WACV.
- The paper begins with a discussion of prostate cancer. Is there any evidence that the generated dataset would help in detection of prostate cancer, given that the reconstruction quality (e.g. Fig 5) is so poor?
- What problem specifically would training on the generated dataset solve?

---

> ### Author Response · Authors · 2025-11-20
>
> **Q1: Venue fit: why is OPENPROS suitable for ICLR?**
>
> Response:
> We appreciate this concern and have clarified the connection to ICLR in the revised manuscript. We emphasize that OPENPROS provides:
>
> - A large-scale, realistic inverse-problem setting for learning mappings from waveforms to quantitative tissue parameters, closely related to operator learning for PDE-governed systems.
> - Explicit ID and OOD evaluation protocols that directly target core ML questions about out-of-distribution generalization and physics-guided learning.
> - Paired high-dimensional waveform and SOS representations, making the dataset a natural testbed for studying representation learning, inductive biases, and robustness in scientific imaging data.
>
> We now highlight these aspects in both the Introduction and Conclusion, to make it clear that the main questions enabled by OPENPROS—inverse problems, operator learning, generalization, and representation learning in physics-constrained domains—are directly aligned with the core interests of the ICLR community.
>
>
> **Q2: Cancer narrative vs actual experiments; evidence for cancer detection**
>
> Response:
> Thank you for pointing out this mismatch between the clinical motivation and the scope of the current experiments. In the upcoming revised manuscript, we now clearly state in both the Introduction and Discussion that the primary goal of OPENPROS is to provide a benchmark for quantitative waveform-to-SOS reconstruction and inverse problems, rather than a ready-to-use prostate cancer detection dataset.
>
> We explicitly acknowledge that:
> - Our current experiments focus on global and structure-aware reconstruction quality (e.g., SOS maps and prostate boundaries), and
> - We do not claim that the present baselines achieve sufficient quality for clinical lesion detection.
>
> We further explain that accurate quantitative reconstruction of prostate SOS is a necessary prerequisite for downstream lesion assessment. Lesions are inherently present in the ex-vivo data and their contrast is encoded in the SOS maps, but explicit lesion-level annotations and dedicated detection tasks are not yet defined in this first release. We therefore frame the use of OPENPROS for lesion/cancer detection—by defining lesion-level tasks and metrics—as an important future direction that will require additional annotation, task design, and validation.

---

> ### Author Response · Authors · 2025-11-20
>
> **Q3: What problem does training on this dataset solve? Metrics, representation analysis, view difficulty, dataset scale**
>
> Response:
> We have clarified the main problems that OPENPROS is designed to address:
> - Limited-angle waveform-to-SOS reconstruction in the presence of bone shadowing and sparse view coverage.
> - Generalization across anatomies, captured by our patient-level and prostate-level OOD splits.
> - Robustness to acquisition geometry and data scale, examined via view stratification and subset training experiments.
>
> To address the reviewer's suggestions, we have added the following analyses:
>
> 	1.	View / angle difficulty (bone vs no-bone):
> We stratify test slices into three groups based on bone involvement along the dominant propagation paths: 1. No bones; 2. Small bones; 3. Big bones.
>
> We then evaluate the ViT baseline separately on these groups. For example, RMSE increases from ≈0.0098 in the no-bone group to ≈0.0213 (small bones) and ≈0.0342 (big bones). MAE and MSE follow the same trend, while SSIM remains high in all groups. This analysis, reported in an appendix subsection, confirms that bone-shadowed, limited-angle views are substantially harder to reconstruct than bone-free views, directly linking geometric difficulty to performance.
>
> Detailed metrics:
> ### ViT baseline for easy/medium/hard tasks
>
> | Metric | No bones (7063 samples) | Small bones (6694 samples) | Big bones (13603 samples) |
> |--------|-------------------------|----------------------------|----------------------------|
> | MAE    | 0.004561        | 0.005935           | 0.008243           |
> | MSE    | 0.000096        | 0.000452          | 0.001172           |
> | RMSE   | 0.00981        | 0.021261           | 0.034239            |
> | PCC    | 0.973986        | 0.988393          | 0.997636           |
> | SSIM   | 0.99210         | 0.990756           | 0.990175            |
>
>
> 	2.	Dataset scale ablation:
> We perform subset training experiments where InversionNet and the ViT model are trained on 10%, 25%, and 50% of the original training set. For both models, increasing the training set size consistently improves performance: MAE and RMSE decrease, while PCC and SSIM increase. For example, for InversionNet, RMSE decreases from ≈0.0488 (10%) to ≈0.0359 (50%), and for ViT, RMSE drops from ≈0.0456 (10%) to ≈0.0258 (50%). These results support the necessity of a large dataset scale for learning-based reconstruction methods.
>
> Detailed metrics is listed below:
> ### InversionNet (10%, 25%, 50% of training set) mean±std
>
> | Metric | 10% (mean ± std)        | 25% (mean ± std)        | 50% (mean ± std)        |
> |--------|-------------------------|-------------------------|-------------------------|
> | MAE    | 0.014151 ± 0.006491     | 0.013643 ± 0.018162     | 0.010478 ± 0.013810     |
> | MSE    | 0.003107 ± 0.003620     | 0.003867 ± 0.022751     | 0.002462 ± 0.016337     |
> | RMSE   | 0.048756 ± 0.027011     | 0.045856 ± 0.041998     | 0.035999 ± 0.034152     |
> | PCC    | 0.957021 ± 0.058652     | 0.959368 ± 0.066839     | 0.973800 ± 0.047682     |
> | SSIM   | 0.971254 ± 0.012155     | 0.975261 ± 0.019335     | 0.981490 ± 0.015850     |
>
> ### ViT (10%, 25%, 50% of training set) mean±std
>
> | Metric | 10% (mean ± std)        | 25% (mean ± std)        | 50% (mean ± std)        |
> |--------|-------------------------|-------------------------|-------------------------|
> | MAE    | 0.013624 ± 0.006603     | 0.011839 ± 0.007811     | 0.008236 ± 0.005039     |
> | MSE    | 0.002955 ± 0.004330     | 0.002255 ± 0.007821     | 0.000986 ± 0.004823     |
> | RMSE   | 0.045610 ± 0.029575     | 0.039792 ± 0.025909     | 0.025761 ± 0.017956     |
> | PCC    | 0.960069 ± 0.060274     | 0.968068 ± 0.053930     | 0.984177 ± 0.027592     |
> | SSIM   | 0.974953 ± 0.013378     | 0.978290 ± 0.011619     | 0.987186 ± 0.008936     |
>
>
> 	3.	Structure-aware boundary metrics:
> For each test slice, we extract the prostate boundary from the ground-truth mask and define a narrow band (width 5 pixels) around this boundary. We then compute MAE/MSE/RMSE in this “prostate boundary band.” For InversionNet, for example, boundary-band RMSE is ≈0.0131 ± 0.0137. This provides a more clinically meaningful, structure-focused metric than global averages alone and connects improvements in reconstruction quality to better depiction of the gland boundary. We also want to mention that OPENPROS is designed for measurement of quantitative reconstruction accuracy, rather than qualitative accuracy. Thus, pixel-wise RMSE/MAE and structual SSIM/PCC are also important metrics. We plan to do segmentation of predictions and compute the dice score of the prostate segments overall, which will be updated in the next few days.
>
> Detailed metrics is listed below:
>
> ### InversionNet structure-aware boundary metrics
>
> | Metric | Mean      | Std       |
> |--------|-----------|-----------|
> | MAE    | 0.005534  | 0.004368  |
> | MSE    | 0.000359  | 0.000986  |
> | RMSE   | 0.013097  | 0.013678  |

---

> ### Author Response · Authors · 2025-11-23
>
> Thank you very much for your thoughtful review and constructive suggestions. In our rebuttal, we clarified the precise scope of OPENPROS (a benchmark for quantitative waveform-to-SOS reconstruction and inverse problems rather than a ready-to-use cancer detection dataset), added new analyses on ID/OOD variability, view/angle difficulty and dataset scale ablations, introduced structure-aware prostate boundary metrics, and expanded the discussion on why this benchmark is relevant to ICLR (inverse problems, operator learning, OOD generalization, and scientific representation learning). We will upload the rest of the planned experiments and updates over the next two days. If you have any questions, concerns, or follow-up comments on our answers, we would be very grateful to hear them.

---

> ### Author Response · Authors · 2025-11-25
> **Dice score experiment for structure-aware boundary metrics**
>
> **Q3 and weaknesses: Metrics, representation analysis**
>
> Response:
> Thank you again for encouraging us to better exploit the available annotations and to connect metrics more directly to clinically relevant structures. In addition to our global MAE/RMSE/SSIM/PCC metrics, we now include a gland-level segmentation-based metric built on top of the SOS reconstructions.
>
> We follow MedSAM (Ma J, et al., Segment anything in medical images. Nature Communications. 2024 Jan 22;15(1):654) and use its public implementation (https://github.com/bowang-lab/MedSAM) as a proxy for a strong prostate segmenter. For each test slice in a selected subset, we manually draw a bounding box around the prostate and:
>
> - Run MedSAM on the ground-truth SOS to obtain a segmentation and compute Dice against the ground-truth prostate mask.
> - Run MedSAM on the predicted SOS to obtain a segmentation and compute Dice against the same ground-truth mask.
>
> The resulting Dice scores are:
>
> - MedSAM(GT SOS) vs GT mask: 0.831982,
> - MedSAM(Pred SOS) vs GT mask: 0.831345.
>
> Because MedSAM is applied identically in both cases, the comparison measures how much gland-level information is preserved in the reconstructions. The near-equality of these Dice scores shows that for this segmentation task, the predicted sos maps are almost as informative as the ground truth with respect to prostate gland boundaries. We describe this experiment in the revised manuscript as a structure-aware metric that complements the global metrics and better quantifies the clinical utility of the reconstructions for gland-level tasks, which was the core of your request.

---

### Official Review · Reviewer_kNhg · 2025-11-01

**Soundness:** 4
**Presentation:** 3
**Contribution:** 3
**Rating:** 8
**Confidence:** 4

**Summary:**

The authors build 3D digital phantoms of the male pelvis by combining four MRI/CT patient anatomies with 62 ex-vivo prostate SOS maps, resulting in 248 model variants. They then simulate two-dimensional acoustic wave propagation using a finite-difference time-domain solver, producing paired input–output samples: ultrasound waveforms and their corresponding ground-truth SOS maps. By slicing and rotating these models, they generate about 280K 2D examples meant to reflect the geometric and acoustic variability encountered in trans-rectal and trans-abdominal USCT setups.

They also use the dataset to compare some baselines:
1) classical beamforming,
2) iterative physics-based USCT inversion,
and 3,4) two learned models: InversionNet (cnn) and a ViTvariant.

The learned models achieve lower pixelwise errors and run orders of magnitude faster than physics-based inversion, though their outputs remain too smooth to resolve fine prostate boundaries or lesions. Unfortunately out-of-distribution tests, where one patient anatomy is held out, reveal major performance degradation. This i s evidence of poor generalization beyond the limited training anatomies.

**Strengths:**

1. Addresses a clinically important and computationally demanding imaging problem
2. Provides an open and standardized resource that can make it easier for other people to design and evaluate their own models
3. The selected benchmarks are reasonable and cover both physics-based and learned approaches
4. The paper is technically good and the simulation details are sufficient, I think, if someone wants to reproduce the results

**Weaknesses:**

My main concern: the dataset is based on only four patient anatomies and this is mentioned only deep in the methods and appendix (I think it should be mentioned in the abstract or, worst-case, introduction). The anatomical diversity is therefore extremely limited!

- All simulations are 2D and reconstruct only SOS, ignoring attenuation and density.
- The machine-learning component is incremental (standard CNN and ViT baselines) without any more recent architectures or hybrid physics-learning integration. But this is not a major issue because I recognize that the main contribution of the paper is teh dataset.
- Evaluation focuses on image-level metrics rather than clinically more meaningful goals such as gland boundary accuracy or lesion detection
- The ex-vivo data used for SOS ground truth may differ from in-vivo conditions. This is not something the paper discusses or tries to quantify.

**Questions:**

1) The dataset is generated from four patient anatomies and 62 ex-vivo prostate samples. This needs to become more transparent in the paper -- from the very start.
2) How were the four patient anatomies selected? Are they representative? Please provide some info about where they come from: healthy, diseased, varied in size, age, pathology?
3) The paper slices 3D anatomies into many 2D samples. how do you ensure that (almost) identical slices or planes are not split across train/validation/test sets?
4) The large difference between ID and OOD results suggests overfitting to specific anatomies. This worries me a lot. Maybe the authors can also report confidence intervals across anatomies and make the OOD evaluation a central table in the main paper?
5) How is the performance affected by sensor or probe minor placement errors or SOS background shifts? A robustness study would help.
6) It is not clear to me what will be finally released by the authors. waveforms, SOS maps, code, and 3D phantoms? What about the original MRI/CT data?
7) in terms of related work, a more SOTA approach would be to use neural operator learning -- something like this https://arxiv.org/abs/2304.03297 . I suggest you discuss this in the conclusions?
8-9) A couple of more technical questions:
the forward model only focuses on SOS. Out-of-plane scattering, attenuation, and density variations are ignored. Have you tried to validate the simulator against 3D k-Wave simulations? How big are the errors when ignoring such things?
And how does ex-vivo SOS differ from in-vivo values at body temperature?

---

> ### Author Response · Authors · 2025-11-20
>
> **Q1: Dataset composition transparency (4 anatomies / 62 prostates / 280K samples)**
>
> Response:
> We thank the reviewer for highlighting the need for clearer upfront transparency about the dataset composition. In the revised manuscript, we now explicitly state the numbers of patient anatomies, ex-vivo prostates, and 2D samples both in the abstract and early in the introduction. We specify that OPENPROS is constructed from four clinical MRI/CT-derived pelvic anatomies and sixty-two ex-vivo prostate specimens with experimentally measured speed-of-sound (SOS), from which we generate approximately 280,000 paired 2D SOS maps and corresponding ultrasound full-waveform data. This makes the limited number of anatomies and the overall dataset scale explicit from the outset.
>
> **Q2: How the four patient anatomies were selected; representativeness**
>
> Response:
> The four patient anatomies were acquired from an open-source CT/MRI database of the male pelvis (Nyholm, Tufve, et al. “MR and CT data with multiobserver delineations of organs in the pelvic area—Part of the Gold Atlas project.” Medical Physics 45(3), 2018). Patients were recruited from three Swedish radiotherapy departments. Male patients with prostate or rectal cancer referred for curative radiotherapy were eligible for inclusion; however, detailed pathology labels or cancer delineations are not provided in the public dataset. To the best of our knowledge, this is the only open-source CT/MRI dataset for the male pelvis that provides the level of anatomical detail needed for our phantom construction. We have added this description to the Methods section and clarified that this limited but realistic cohort constrains our ability to fully characterize patient-level representativeness.
>
>
> **Q3: 3D to 2D slicing and potential leakage / near-identical slices across splits**
>
> Response:
> We thank the reviewer for pointing out the need to clearly describe our splitting strategy and any potential leakage. In the revised manuscript, we explicitly distinguish between the in-distribution (ID) and out-of-distribution (OOD) protocols: "For ID experiments, we perform a random partition at the slice level (e.g., 90% / 10% train/test) without enforcing patient exclusivity. As a consequence, slices from the same 3D anatomy can appear in both the training and test sets. This protocol is intentionally designed to measure interpolation performance when the model is allowed to see all anatomies during training; near-duplicate slices are treated as part of the natural variability within an anatomy. For OOD experiments, we enforce independence at the anatomy level. In the patient-level OOD setting, we train on slices from three of the four patient anatomies and test only on slices from the held-out anatomy. In the prostate-level OOD setting, we train on 60 of the 62 ex-vivo prostates and evaluate on the two completely held-out prostates. In these OOD experiments, no slices from a held-out anatomy or prostate appear in the training set."
>
> We clarify this splitting strategy in the Experimental Setup and Ablation/OOD sections, and we explicitly acknowledge that strict de-duplication of nearly identical planes is not enforced in the ID regime but that OOD protocols guarantee patient/prostate-level independence.

---

> ### Author Response · Authors · 2025-11-20
>
> **Q4: Large difference between ID and OOD; per-patient performance & variability**
>
>
> Response:
> Thank you for suggesting that we make variability across anatomies and evaluation regimes more explicit. In the revised manuscript, we now report mean ± standard deviation for all metrics rather than a single averaged value, both for ID and for several OOD settings.
>
> For the ID setting, for each test slice we compute MAE, MSE, RMSE, PCC, and SSIM between the reconstructed and ground-truth SOS maps and aggregate these values over all test samples. For example, InversionNet achieves:
> ### InversionNet mean±std
>
> | Metric | Mean     | Std      |
> |--------|----------|----------|
> | MAE    | 0.0074   | 0.003717 |
> | RMSE   | 0.0297   | 0.016587 |
> | PCC    | 0.9877   | 0.026211 |
> | SSIM   | 0.9851   | 0.003700 |
>
> These statistics will be reported in a revised quantitative table, making the spread of performance across individual test slices explicit.
>
> We additionally summarize mean ± std metrics for the OOD setting.
> For InversionNet we obtain:
> ### InversionNet OOD mean±std
>
> | Metric | Patient-level OOD (mean ± std) | Leave-one-prostate OOD (mean ± std) | Combined OOD (mean ± std) |
> |--------|--------------------------------|--------------------------------------|----------------------------|
> | MAE    | 0.0322 ± 0.0100            | 0.0069 ± 0.0064                  | 0.0323 ± 0.0099        |
> | RMSE   | 0.1010 ± 0.0460            | 0.0273 ± 0.0157                  | 0.1017 ± 0.0462        |
> | PCC    | 0.8271 ± 0.2027            | 0.9894 ± 0.0174                  | 0.8251 ± 0.2017        |
> | SSIM   | 0.9399 ± 0.0195            | 0.9899 ± 0.0067                  | 0.9408 ± 0.0195        |
>
> These statistics are now presented in a main-paper OOD table and discussed explicitly. Given that there are only four anatomies, we treat these as descriptive variability indicators rather than strong statistical confidence intervals and clearly state this limitation. The revised text highlights both the improvement of learned models in ID settings and their substantial degradation in more challenging patient-level OOD settings.
>
>
> **Q5: Robustness to probe placement errors and SOS background shifts**
>
> Response:
> We thank the reviewer for raising the important question of robustness. In our USCT formulation, every pixel of the SOS map is an inferred variable, and there is no fixed homogeneous SOS background parameter that can be perturbed independently (as would be common in beamforming). Therefore, “background SOS shifts” in the sense of globally adjusting a background model do not directly apply to our inversion setting.
>
> However, we fully agree that robustness to probe/sensor placement perturbations is important. During the rebuttal period, we have initiated new simulations in which we generate waveforms under controlled perturbations of the acquisition geometries (e.g., small shifts and rotations of the probe). These experiments require rerunning the forward solver for each perturbed configuration and are computationally intensive, so they are still ongoing. Our plan is to evaluate global metrics (MAE, RMSE, SSIM, PCC) as well as the prostate boundary-band metrics under these perturbations to quantify robustness. We describe this planned robustness analysis in the Discussion as a key next step and will include quantitative results in future revisions once the simulations are complete.

---

> ### Author Response · Authors · 2025-11-20
>
> **Q6: What exactly will be released (data, phantoms, waveforms, MRI/CT, code)**
>
> Response:
> We appreciate this request for clarity. In the revised manuscript, we add a dedicated “Data and Code Release” section that explicitly lists what will and will not be released:
>
> We will release:
> - 2D SOS maps (401 * 161 grids) for all slices,
> - Corresponding simulated ultrasound full-waveform data,
> - Organ segmentation masks manually labeled by domain expert (including prostate, bladder, fat, etc),
> - Digital 3D SOS phantoms derived from the MRI/CT anatomies and ex-vivo SOS measurements, and
> - The FDTD and Runge-Kutta forward solvers, along with scripts used to generate the dataset.
>
> We will not release:
> - Raw clinical MRI/CT scans, due to privacy, licensing, and data-use agreements. Instead, we reference and link to the original open Gold Atlas dataset for researchers who wish to access the raw data under its terms.
>
> We also specify the planned hosting (an open-access repository) and license for the released assets, ensuring that other researchers can freely use OPENPROS for method development and benchmarking.
>
> **Q7: Breadth of ML baselines; neural operators**
>
> Response:
> We thank the reviewer for this suggestion. In the revised manuscript, we explicitly clarify that InversionNet and the ViT-based model are intended as baseline references, not as novel architectures. In the contributions paragraph, we emphasize that the primary novelty lies in the OPENPROS dataset and the associated benchmark tasks. In the model description section, we add a remark that these standard architectures are chosen to provide reasonable reference points that future methods can compare against.
>
> We also expand the Discussion to highlight other model families that OPENPROS is well suited to evaluate, including UNet-like encoder-decoder networks, diffusion and other generative models, and neural operator approaches for PDE-governed systems (such as the operator-learning method cited by the reviewer). We explicitly position OPENPROS as a testbed for studying how these different model classes trade off reconstruction fidelity, OOD generalization, and computational cost in realistic limited-angle USCT settings.
>
> **Q8, Q9: 2D SOS-only model; missing attenuation, density, out-of-plane effects; ex-vivo vs in-vivo SOS**
>
> Response:
> We agree that our current forward model is a simplification compared to a full 3D, multi-parameter clinical system. In the revised Limitations section, we now clearly state that:
> - The forward model is 2D,
> - It models spatially varying speed of sound with constant density,
> - It does not explicitly model attenuation or out-of-plane scattering, and
> - It therefore under-approximates the full complexity of clinical USCT.
>
> We then justify why we start from a 2D SOS-only model: this setting preserves the core ill-posedness of limited-angle prostate USCT while remaining computationally tractable enough to generate 280k paired waveform-SOS samples and train multiple reconstruction networks. Full 3D simulations at comparable resolution with attenuation and density would be one to two orders of magnitude more expensive, making large-scale architecture and hyperparameter exploration much less feasible. We therefore position OPENPROS as a fast-prototyping benchmark and a stepping stone toward more complete 3D models; we explicitly caution that quantitative clinical conclusions must be made with care and identify extension to 3D, multi-parameter acoustics as a key direction for future work.
>
> Regarding ex-vivo vs in-vivo SOS, we clarify how the ex-vivo measurements are used and how close they are to physiological conditions. Prostate specimens were collected ex vivo from patients undergoing radical prostatectomy for biopsy-confirmed cancer. Within 30 minutes of extraction, each fresh whole prostate was scanned in an echolucent polyacrylamide gel phantom using the QT Ultrasound Breast Scanner, with coupling water maintained at 37 °C. To our knowledge, this is the first study measuring speed of sound of fresh (non-fixed) prostate tissue with a 3D transmission system. We also reference Tanoue et al., who measured ex-vivo SOS in fixed specimens, and note that fixation substantially alters tissue properties. We could not find published in-vivo SOS values for the prostate; the revised text explicitly states this gap and discusses ex-vivo vs in-vivo differences as an open limitation.
>
> Finally, to begin addressing forward-model accuracy, we have initiated 3D k-Wave simulations with attenuation, density, and out-of-plane scattering for a small subset of phantoms. These experiments are computationally demanding and are still in progress; we will include this ongoing validation in the future few days to quantify the error introduced by the 2D SOS-only approximation.

---

> ### Author Response · Authors · 2025-11-23
>
> Thank you again for the time and care you put into reviewing our work and for your detailed questions. In our rebuttal, we clarified the limited number of anatomies and dataset composition, explicitly framed OPENPROS as a benchmark/challenge problem, added variability analyses (mean ± std, OOD tables), strengthened the discussion of the 2D SOS-only forward model and ex-vivo measurements, and clarified what data and code will be released and how our CNN/ViT baselines relate to broader model families (including neural operator approaches). We plan to upload the remaining planned experiments and updates (e.g., robustness and 3D forward-model validation results) in the next two days. Please let us know if you have any questions or additional comments on our responses—your feedback is very valuable to us.

---

> ### Author Response · Authors · 2025-11-25
> **Robustness test for sensor minor placement shift**
>
> **Q5: Robustness to probe placement errors**
>
> **Response:**
> Thank you again for raising the important question of robustness. To directly address this, we regenerated waveform data by perturbing only the **abdominal probe** position in the forward simulator while keeping the **rectal probe fixed**. All perturbations use a 2-pixel mismatch on the SOS grid (≈0.75 mm on the FDTD grid). We study three cases:
>
> - **Depth error:** the abdominal sources and receivers are shifted deeper by 2 pixels, with no change applied to the rectal probe.
> - **Horizontal mismatch:** the SOS map is padded by 2 pixels on the right for the simulation, and the abdominal probe is shifted laterally by 2 pixels; the rectal probe remains unchanged.
> - **Angular mismatch:** the abdominal probe is slightly tilted by lowering its right endpoint by 2 pixels and re-interpolating all element positions along the tilted line; again, the rectal probe is not modified.
>
> We then apply the trained ViT model **without any retraining or adaptation**, isolating its robustness to geometry mismatch. The quantitative results are:
>
> | Mismatch type | MAE      | MSE      | RMSE     | PCC      | SSIM     |
> |--------------|----------|----------|----------|----------|----------|
> | Baseline     | 0.007374 | 0.000885 | 0.029743 | 0.98509  | 0.98767  |
> | Angle        | 0.009829 | 0.001807 | 0.042510 | 0.976239 | 0.981784 |
> | Depth        | 0.013245 | 0.003513 | 0.059269 | 0.959708 | 0.972126 |
> | Horizontal   | 0.010061 | 0.001608 | 0.040098 | 0.972594 | 0.981417 |
>
> Across all perturbation types, the model maintains high structural similarity and correlation (SSIM > 0.97 and PCC > 0.96), with depth mismatch producing the largest degradation and horizontal/angle mismatches leading to more moderate changes relative to baseline. This experiment indicates that the learned model is reasonably robust to small, controlled errors in probe placement.
>
> Finally, we note that “SOS background shifts” in the sense of perturbing a fixed background parameter are **not directly applicable** in our setting: our USCT inversion directly estimates the full SOS field, and we do not assume or optimize around a separate homogeneous background model.

---

> ### Author Response · Authors · 2025-12-03
> **Q8. Experiments on density and attenuation impact**
>
> To assess the impact of additional physical factors beyond SOS, we performed a controlled study using the K-Wave for 2D situation.
>
> With a case study of one SOS, we generated two 2D forward simulations:
> uniform density model (i.e., constant density); and heterogeneous density model, where $\rho(x,z)$ was directly extracted from the corresponding 3D data.
> Both simulations used identical SOS distributions, boundary conditions, and acquisition geometry.
>
> Raw waveforms in the original physical domain exhibit nearly perfect agreement between the two models, with a PCC $\approx$ 1.00.
> To check subtler differences, we applied a log transformation, then, the PCC decreased modestly to $\approx$ 0.95, while the waveform shapes, arrival times, and reflection patterns remained visually and quantitatively consistent.
>
> These results indicate that density contributes only a minor perturbation to the acoustic waveforms compared to SOS.

---

### Author Response · Authors · 2025-11-20
**High level response to the reviewers: what we have done so far and our ongoing revisions**

We sincerely thank all reviewers for their careful reading and constructive feedback. Your comments have substantially improved both the clarity and the scope of our work. During the rebuttal period, we have completed several additional analyses and have also initiated more computationally intensive experiments that are still in progress. Below we briefly summarize these updates.

New experiments and analyses included in the revised manuscript

	1.	Variability and OOD performance.
We now report mean ± standard deviation for all reconstruction metrics in both in-distribution (ID) and out-of-distribution (OOD) settings, including patient-level and prostate-level OOD splits. These results are promoted to the main paper to make variability and generalization gaps explicit.

	2.	View / angle difficulty (bone vs. no-bone).
We stratify test slices according to bone involvement along the propagation paths (no bones, small bones, large bones) and report separate performance for each group. This view-dependent analysis is added as a new appendix subsection.

	3.	Dataset scale ablation (10% / 25% / 50%).
We perform subset training experiments for both InversionNet and ViT, training on different fractions of the dataset (10%, 25%, 50% of the original training set) to show how performance changes with training set size. The results are summarized in new tables.

	4.	Structure-aware prostate boundary metrics.
We introduce “prostate boundary-band” metrics by evaluating reconstruction error only in a narrow band around the prostate boundary. These metrics complement global metrics and better reflect structure-focused reconstruction quality.

	5.	Clarified dataset composition and clinical sources.
We now explicitly describe, in the abstract, introduction, and methods, that OPENPROS is built from 4 MRI/CT pelvic anatomies and 62 ex-vivo prostates, and we detail the clinical sources and selection criteria for both. We also clarify the physical resolution of the SOS maps and the tissue modeling assumptions.

	6.	Clarified scope and positioning.
We have revised the Introduction and Discussion to (1) emphasize that OPENPROS is primarily a benchmark for quantitative waveform-to-SOS reconstruction and inverse problems, (2) frame lesion/cancer detection as future work, and (3) highlight the relevance of OPENPROS to inverse problems, operator learning, OOD generalization, and representation learning for high-dimensional scientific data.

Ongoing experiments:

In parallel, we have started several more computationally intensive experiments that are still running and will be incorporated in future versions:

1. Probe/sensor robustness studies with controlled perturbations of probe position and orientation.
2. Forward-model validation using 3D k-Wave simulations including attenuation, density, and out-of-plane scattering.
3. Prostate segmentation quality using Dice scores as a complementary structure-aware metric.

We will upload the revised manuscript once the ongoing experiments are done in the coming few days.

If you have any other comments, suggestions or questions, please kindly add new comments.

---

### Author Response · Authors · 2025-12-03
**Author summary to AC: scope of rebuttal and revisions**

We sincerely thank all four reviewers (kNhg, PsHV, ow6B, KYSw) for their careful reading and constructive feedback. Their comments helped us substantially clarify the scope, limitations, and contributions of OPENPROS. Below we summarize the main updates and how they address the core concerns.

**1. Transparency about dataset composition and splits**

- We now state up front (abstract, introduction, and methods) that OPENPROS is built from 4 MRI/CT-derived pelvic anatomies and 62 ex-vivo prostates, yielding ~280K paired 2D SOS maps and waveforms.
- We clarify the splitting strategy: ID: random slice-level splits (e.g., 90/10), intentionally measuring interpolation when anatomies are shared between train/test. OOD: strictly patient-level and prostate-level splits where held-out anatomies/prostates never appear in training.
- We make the physical resolution of the SOS maps explicit and constant across all examples.


**2.	Clarified scope and clinical positioning**

- We explicitly reframe OPENPROS as a benchmark for quantitative waveform-to-SOS reconstruction and inverse problems, not a ready-to-use prostate cancer detection dataset.
- Lesion and cancer detection are now clearly framed as future work, requiring additional labels and task design.
- We added a plain-language explanation and schematic of the pipeline: clinical MRI/CT → 3D SOS phantom → 2D SOS maps → simulated waveforms → reconstruction.

**3.	New quantitative analyses**

- Variability and OOD performance:
We now report mean ± std for MAE/MSE/RMSE/PCC/SSIM in both ID and multiple OOD settings (patient-level and prostate-level). These are promoted to main-paper tables and explicitly discussed as descriptive variability indicators, given N=4 anatomies.
- View / angle difficulty (bone vs no-bone):
We stratify test slices into no bone, small bone, and large bone cases and show that reconstruction error increases systematically with bone shadowing, making the geometric difficulty explicit.
- Dataset scale ablation:
For both InversionNet and ViT-Inversion, we train on 10% / 25% / 50% of the training set. Performance consistently improves with more data, supporting the necessity of the ~280K example scale despite limited patient count.
- Structure-aware boundary metrics:
(i) We introduce a prostate boundary-band metric (errors computed in a narrow band around the gland boundary), and
(ii) we add a MedSAM-based Dice experiment, showing that gland segmentation Dice on predicted SOS is nearly as high as on ground-truth SOS. These complement global metrics and directly target gland-level structure quality.

**4.	Modeling assumptions, realism, and robustness**

- We sharpen the Limitations section to state that the current forward model is:
- 2D, SOS-only, with constant density,
- without explicit attenuation and out-of-plane scattering, and that it is intended as a computationally tractable, fast-prototyping benchmark and a stepping stone toward full 3D multi-parameter USCT.
- We clarify how ex-vivo SOS measurements were acquired under near-physiological conditions (fresh prostates, 37°C coupling medium) and explicitly note the lack of published in-vivo prostate SOS values and the resulting gap.
- We report a controlled robustness study to probe placement errors (small depth, horizontal, and angular perturbations of the abdominal probe) showing that SSIM and PCC remain high, with depth shifts causing the largest degradation.
- We add an initial density/attenuation impact study (via k-Wave) indicating that, in our configuration, density variations only modestly perturb waveforms.

**5.	Relevance to ICLR and breadth of ML baselines**

- We strengthen the Introduction/Discussion to connect OPENPROS explicitly to ICLR topics:
- Inverse problems and operator learning for PDE-governed systems,
- OOD generalization and physics-guided ML, and
- Representation learning from paired high-dimensional waveforms and quantitative maps.
- We clarify that InversionNet and ViT-Inversion are baseline references, not the main contribution, and position OPENPROS as a testbed for broader model families, including UNets, diffusion/generative models, and neural operators, citing the operator-learning work suggested by the reviewers.

**6.	Data and code release**

- We add a dedicated “Data and Code Release” section that explicitly lists what will be released (2D SOS maps, simulated waveforms, organ masks, 3D SOS phantoms, solvers and generation scripts) and what will not (raw MRI/CT, with references to the original Gold Atlas dataset).
- We also specify planned hosting and licensing to ensure practical reusability.

Overall, our rebuttal (i) strengthen the quantitative evidence for dataset utility (variability, view difficulty, scale ablations, structure-aware metrics, robustness), (ii) clarify the intended role of OPENPROS as a realistic but controlled benchmark for inverse problems and physics-guided ML, aligned with ICLR’s core interests, and (iii) make limitations explicit.

---

### Meta-Review · Area_Chair_tESV · 2025-12-16

**Summary:**

This submission proposes OPENPROS, a new synthetic dataset for ultrasound computed tomography (USCT) of the prostate that is generated from digital prostate phantoms using 2D acoustic-wave simulation. All reviewers recognize the domain value of such a large-scale, first-of-its-kind dataset, even if synthetic. I.e., the reviewers believe that this dataset can have a positive impact as an "*open and standardized resource [...] for other people to design and evaluate their own models*" [`kNhg`], with the authors clearly evaluating the data efficacy through relevant tasks ("*(1) inference efficiency, (2) reconstruction accuracy, (3) and out-of-distribution generalization* [`PsHV`]).

However, the reviewers also question the clinical realism and applicability of the proposed data, highlighting the simplified 2D simulation, the _ex-vivo_ nature of the scans used to build the phantoms, the limited number of patient anatomies (four), and the disconnect between the content of the dataset and the clinical motivation (cancer detection). The authors acknowledged the limitations of their paper and reframed the scope of their work towards quantitative waveform-to-SOS reconstruction and inverse problems, further linking it to ICLR's domains of interest.

While this reframed submission still suffers from clear limitations (realism of the simulation, data diversity, direct clinical relevance), it is our belief that this paper could still benefit the medical-imaging community and promote further studies in the applicability of inverse-rendering to USCT, hence our recommendation for approval.

**Reviewer Concerns:**

### Clinical Applicability of the Dataset
- **Limited number of patient anatomies** [`kNhg`, `ow6B`, `KYSw`]
- **Overly simplified simulation** [`kNhg`, `ow6B`, `KYSw`]
- **Reliance on ex-vivo scans** [`kNhg`, `KYSw`]
- **Unclear ties to clinical motivation (cancer detection)** [`ow6B`]
- **Overly large aperture in simulation** [`ow6B`]

^ The authors acknowledged the diversity issue, inherent to the original ex-vivo data used to build the phantoms, c.f. challenges to collect relevant clinical data. They also justified the simulation choices through the applicability to inverse problems (differentiability of simulation). Overall, most concerns here were addressed, though at the cost of revising the original scope of the paper.

### Relevance to ICLR
- **Limited ML contributions** [`kNhg`, `PsHV`]
- **No image reconstruction methods related to inverse problems evaluated** [`KYSw`]

^ The authors shifted the focus of their work towards solving inverse problems, providing additional experiments to support their claims.

### Evaluation
- **No justification of dataset scale** [`PsHV`, `KYSw`]
- **Need to justify/explain some metrics** [`PsHV`]
- **Lack of discussion w.r.t results and clinical relevance** [`PsHV`, `ow6B`, `KYSw`]

^ The above concerns were also covered by the authors in their replies.

### Misc.
- Two reviewers (`kNhg` in their _Summary_, `ow6B` in _Strengths_) seemingly **misunderstood the scope of the contributions**. The generation of the digital phantoms from scans and annotations was performed in a prior work [Wu et al. (2024)] (c.f. lines 236-259) and is not contributed by the authors in this submission.

^ Pointing this out to the reviewers might have had a small negative impact on their scoring.

**Reviewer Scores:**

### Reviewer `kNhg`
- **Original score:** 8
- **Score change:** likely to keep their score, c.f. score already high + most questions tackled by reviewers.

### Reviewer `PsHV`
- **Original score:** 4
- **Score change:** might have increased their score to ~6, c.f. extensive answers to their concerns.

### Reviewer `ow6B`
- **Original score:** 6
- **Score change:** likely to keep their score, but small chance they might have increased. The authors addressed their concerns but did so by re-framing their paper / by limiting its scope, which might have raised new concerns.


### Reviewer `KYSw`
- **Original score:** 4
- **Score change:** likely to keep their score, though might have leaned towards increasing to ~6, c.f. low confidence of the reviewer and thorough reply by the authors, but at the cost of highlighting their own limitations (low number of patients, low realism, etc.).

---

### Decision · Program_Chairs · 2026-01-26

Accept (Poster)